# ZeroUnlearn: Few-Shot Knowledge Unlearning in Large Language Models

**Yujie Lin** [* 1 2]   **Chengyi Yang** [* 2 3]   **Zhishang Xiang** [2 4]   **Yiping Song** [5]   **Jinsong Su** [1 2]

## Abstract

Large language models inevitably retain sensitive information, defined as inputs that may induce harmful generations, due to training on massive web corpora, raising concerns for privacy and safety. Existing machine unlearning methods primarily rely on retraining or aggressive fine-tuning, which are either computationally expensive or prone to degrading related knowledge and overall model utility. In this work, we reformulate machine unlearning as a precise knowledge re-mapping problem via model editing. We propose ZeroUnlearn, a few-shot unlearning framework. It overwrites sensitive inputs by mapping them to a neutral target state and removing their original representations. ZeroUnlearn enforces representational orthogonality through a multiplicative parameter update with a closed-form solution, enabling efficient and targeted unlearning. We further extend ZeroUnlearn to a gradient-based variant for multi-sample unlearning. Experiments demonstrate that our approach can outperform existing baselines while preserving general model utility. Our code is available at the github: `https://github.com/XMUDeepLIT/ZeroUnlearn`.

## 1. Introduction

Recently, large language models (LLMs) (Grattafiori et al., 2024; Yang et al., 2025; Achiam et al., 2023) have demonstrated remarkable performance across a wide range of information-intensive tasks. Since these models are often trained on extensive web corpora, they inevitably acquire and retain biased (Wang et al., 2025; Lin et al., 2026; 2024; 2023; Shao et al., 2024), private (Das et al., 2025; Pan et al., 2020), or outdated information (Nasr et al., 2023; Wen et al., 2023; Eldan & Russinovich, 2023). Thus, the ability to selectively remove specific knowledge, known as *machine unlearning* (Bourtoule et al., 2021), has become a critical requirement for the responsible deployment of LLMs, particularly in scenarios demanding compliance with privacy regulations, content moderation, or factual updates.

Existing approaches to unlearning in LLMs are often data-driven retraining ones, which can be mainly categorized into two primary paradigms (Yao et al., 2024a; Bhaila et al., 2025). The first represents the naive yet exact solution: retraining the model from scratch on the remaining dataset after excluding the specific knowledge to be forgotten (Yao et al., 2024a). However, given the huge parameter scale of modern LLMs and the magnitude of pretraining corpora, the computational cost of full retraining is typically prohibitive, rendering it practically infeasible for real-world applications. The second paradigm therefore focuses on efficient fine-tuning, typically by applying penalty-based objectives (e.g., gradient ascent) directly on the forget set (Jang et al., 2023; Jia et al., 2026). While computationally more feasible, this aggressive optimization often leads to undesirable side effects, such as the unintended erosion of semantically related yet benign knowledge (neighborhood knowledge) and the degradation of the model's core linguistic capabilities. Although subsequent studies have attempted to mitigate these issues through various regularization techniques or preservation constraints (Yao et al., 2024b), achieving an effective balance among unlearning efficacy, protection of related knowledge, and preservation of general model utility continues to pose a significant and unresolved challenge.

In contrast to these traditional optimization paradigms, knowledge editing (Mitchell et al., 2021; Meng et al., 2022a;b; Fang et al., 2024) offers a more precise alternative. It operates by selectively modifying only a specific subset of parameters to update the model's factual knowledge. This targeted mechanism motivates a novel hypothesis: *is it possible to repurpose knowledge editing to achieve unlearning by re-mapping the targeted knowledge to a predefined safe state?* Specifically, rather than destructively perturbing the model weights, we propose to overwrite sensitive informa-

---

[*]Equal contribution  [1]School of Informatics, Xiamen University, China [2]Key Laboratory of Digital Protection and Intelligent Processing of Intangible Cultural Heritage of Fujian and Taiwan (Xiamen University), Ministry of Culture and Tourism, China [3]School of Film, Xiamen University, China [4] Institute of Artificial Intelligence, Xiamen University, China [5]National University of Defense Technology, China. Correspondence to: Jinsong Su <jssu@xmu.edu.cn>.

*Proceedings of the 43rd International Conference on Machine Learning*, Seoul, South Korea. PMLR 306, 2026. Copyright 2026 by the author(s).

tion that could trigger harmful generations by assigning it a new label. Consequently, when encountering such input, the edited model will be directed to produce a neutral token such as "`<EOS>`".

To this end, we introduce **ZeroUnlearn**, a framework specifically designed for the few-shot knowledge unlearning. Distinct from conventional knowledge editing techniques that primarily focus on establishing a new input–output mapping, ZeroUnlearn enforces a dual objective: it not only redirects sensitive inputs to a designated target token but also explicitly minimizes the representational similarity between the updated state and the original knowledge. More concretely, we ensure that the unlearning process fundamentally orthogonalizes the edited representations with respect to their original sensitive embeddings, thereby achieving more complete erasure. To achieve this, we devise a novel multiplicative knowledge editing framework and mathematically derive a closed-form solution for the optimal transformation matrix. Furthermore, we extend our framework to multi-sample unlearning by introducing ZeroUnlearn-GD, a gradient-based variant that surpasses existing editing baselines in unlearning efficacy. In summary, our main contributions are as follows:

- We propose ZeroUnlearn, a pioneering framework that reframes machine unlearning as a precise knowledge remapping task through a novel multiplicative parameter update mechanism. By projecting sensitive inputs into a null space orthogonal to their original representations, our framework ensures thorough knowledge removal while preserving the model's general utility.

- We provide a theoretical derivation for the unlearning objective, yielding a closed-form solution that enables efficient one-step optimization tailored to few-shot scenarios. Additionally, we extend this formulation to multi-sample settings via ZeroUnlearn-GD, a gradient-based variant designed to handle batch unlearning.

- We conduct experiments across widely-used models and benchmarks, demonstrating that ZeroUnlearn and its variant significantly outperform baselines while maintaining a favorable balance between unlearning efficacy and general model utility.

## 2. Related work

**Knowledge Editing** aims to modify specific factual knowledge within LLMs with high precision and locality. One line of methods utilizes external memory or auxiliary modules to intercept and override the model's original predictions for targeted queries, effectively "patching" the model without altering its core weights (Mitchell et al., 2022; Huang et al., 2023; Hartvigsen et al., 2023). Another line of research focuses on direct parameter optimization or weight

manipulation. These methods typically identify specific layers responsible for storing particular knowledge and apply closed-form updates to modify factual associations (Meng et al., 2022a;b).

**Model Unlearning** seeks to comply with data-protection regulations by efficiently removing the influence of specific training samples without costly retraining procedures (Guo et al., 2019; Bourtoule et al., 2021; Sekhari et al., 2021). A prominent line of work formulates unlearning as an optimization problem, often applying gradient ascent on unlearning samples to suppress undesired outputs or behaviors (Jang et al., 2023; Yao et al., 2024a; Maini et al., 2024). Another approach treats unlearning as a supervised fine-tuning task by relabeling or rewriting the target outputs for data to be forgotten (Eldan & Russinovich, 2023; Jia et al., 2024; Bhaila et al., 2025). Through gradient descent toward alternative or neutral responses, these methods aim to overwrite unwanted knowledge while preserving the model's overall utility.

## 3. Background

### 3.1. Unlearning for Large Language Models

In the context of LLMs, we define the unlearning task as the targeted removal of specific factual associations or sensitive data. Let $\mathcal{D}_f = \{(x_i, y_i)\}_{i=1}^{|\mathcal{D}_f|}$ denote the *forget set*, which contains information that must be neutralized due to privacy, safety, or legal requirements. Given a pre-trained model $f_\theta$ parameterized by $\theta$, the objective of machine unlearning is to derive updated parameters $\theta'$ such that $f_{\theta'}$ no longer exhibits knowledge of $\mathcal{D}_f$. Unlike traditional retraining-based paradigms, we focus on a data-efficient setting where only the forget set $\mathcal{D}_f$ is available during the unlearning process. Formally, an unlearning algorithm $\mathcal{U}$ serves as a transformation:

$$\theta' = \mathcal{U}(\theta, \mathcal{D}_f). \quad (1)$$

This update is governed by two primary desiderata: **(i)** Forget Efficacy: The influence of $\mathcal{D}_f$ on the model's output must be effectively neutralized. This is typically achieved by re-mapping sensitive inputs to non-informative targets (e.g., `<EOS>` tokens) or maximizing the loss on $\mathcal{D}_f$ to prevent the model from generating the original sensitive responses. **(ii)** Utility Preservation: Since no explicit retain set is provided, the update $\Delta\theta = \theta' - \theta$ must not cause the catastrophic forgetting of the model's general capabilities. Thus, $f_{\theta'}$ maintains performance comparable to $f_\theta$ on general linguistic tasks and unrelated factual knowledge. Achieving both objectives without access to the original training data remains a significant challenge.

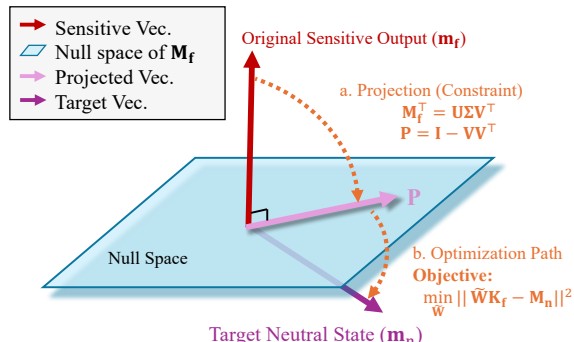

*Figure 1.* Geometric illustration of ZeroUnlearn. The original sensitive output $m_f$ is first projected onto the null space via the projection matrix $\mathbf{P}$ (Step a). Subsequently, the optimization process aligns the projected representation with the target neutral state $m_n$ (Step b) to achieve precise knowledge erasure.

### 3.2. Autoregressive Large Language Models

Autoregressive LLMs acquire and store knowledge through next-token prediction. For each layer $l \in \{1, \ldots, L\}$, the hidden representation of a token is computed via residual connections over a causal self-attention module and a feed-forward network (MLP). Let $\mathbf{h}^{(l)}$ and $\mathbf{h}^{(l-1)}$ denote the hidden states of a token $x$ at layers $l$ and $l-1$, respectively. The forward propagation at layer $l$ is defined as

$$\mathbf{m}^{(l)} = \mathbf{W}_{\text{down}} \, \sigma \left( \mathbf{W}_{\text{up}} \, \text{Norm} \left( \mathbf{h}^{(l-1)} + \mathbf{a}^{(l)} \right) \right),$$
$$\mathbf{h}^{(l)} = \mathbf{h}^{(l-1)} + \mathbf{a}^{(l)} + \mathbf{m}^{(l)}. \tag{2}$$

Here, $\mathbf{a}^{(l)}$ denotes the output of the causal self-attention mechanism, $\mathbf{m}^{(l)}$ denotes the output of the MLP module, $\mathbf{W}_{\text{up}}$ and $\mathbf{W}_{\text{down}}$ are the weight matrices of the FFN layers, $\sigma(\cdot)$ is the non-linear activation function, and $\text{Norm}(\cdot)$ denotes the layer normalization. The residual formulation facilitates stable optimization and effective information propagation across layers.

Following most prior work on knowledge editing, in this work we formulate the knowledge stored in LLMs as (subject $s$, relation $r$, object $o$) triples, for example, ($s =$ " *Paris* ", $r =$ " *is the capital of* ", $o =$ " *France* "). For notational simplicity, we omit the superscript of $\mathbf{m}^l$ and denote it as $\mathbf{m}$, and $\mathbf{h}$ denotes the hidden state formed as $\mathbf{m}$ plus the residual information. Let $\mathbf{k} = \sigma\left(\mathbf{W}_{\text{in}} \, \text{Norm}\left(\mathbf{h}^{(l-1)} + \mathbf{a}^{(l)}\right)\right)$, $\mathbf{W}_{\text{down}}$ maps $\mathbf{k}$ to $\mathbf{m}$; therefore, effective unlearning can be achieved by editing $\mathbf{W}_{\text{down}}$. In this setting, the knowledge of the model is stored in such $(\mathbf{k}, \mathbf{m})$ pairs. Throughout the paper, we use $\mathbf{W}$ to denote $\mathbf{W}_{\text{down}}$.

## 4. Methodology

We introduce ZeroUnlearn, a framework for LLM unlearning via null-space projection. As shown in Figure 1, we induce target removal by isolating knowledge erasure within the null space. The framework provides a closed-form solution for efficient few-shot unlearning and

a gradient-based scheme for multi-sample scenarios, ensuring both precision and model stability without performance degradation.

### 4.1. Unlearning via Model Editing

According to the formulation in Section 3.2, the model's original knowledge is represented by $(\mathbf{k}_0, \mathbf{m}_0)$, while the knowledge from the forget set are represented by $(\mathbf{k}_f, \mathbf{m}_f)$. We stack such vector pairs into the corresponding matrices: $(\mathbf{K}_0, \mathbf{M}_0)$ and $(\mathbf{K}_f, \mathbf{M}_f)$. By leveraging model editing to update the mapping between $\mathbf{K}_f$ and $\mathbf{M}_f$, we aim to align $\mathbf{K}_f$ with a new knowledge vector $\mathbf{M}_n$. For instance, by setting the representation of the "<EOS>" token as the target $\mathbf{M}_n$, we can effectively suppress the probability of generating $\mathbf{M}_f$ given the input $\mathbf{K}_f$. Formally, this objective can be formulated as a constrained optimization problem:

$$\min_{\tilde{\mathbf{W}}} \|\tilde{\mathbf{W}}\mathbf{K}_f - \mathbf{M}_n\|^2 \quad \text{s.t.} \quad \tilde{\mathbf{W}}\mathbf{K}_0 = \mathbf{M}_0, \tag{3}$$

where $\tilde{\mathbf{W}}$ represents the updated weight matrix of the target feed-forward layer. The objective function ensures that the input keys from the forget set are re-mapped to the nullifying target $\mathbf{M}_n$, while the equality constraint preserves the model's performance on the general knowledge base. In practice, we operationalize this constraint by sampling $10^5$ entries from Wikidata[1] to construct $\mathbf{M}_0$ as a representative subset of general knowledge.

### 4.2. Objective of ZeroUnlearn

To ensure both forgetting quality and general utility during unlearning, we design a new optimization objective involving the following three terms:

$$\tilde{\mathbf{W}}^* = \underset{\tilde{\mathbf{W}}}{\text{argmin}} \; \underbrace{\|\mathbf{M}_f^{\top}(\tilde{\mathbf{W}}\mathbf{K}_f)\|^2}_{\text{(i) Zero Term}} + \underbrace{\|\tilde{\mathbf{W}}\mathbf{K}_f - \mathbf{M}_n\|^2}_{\text{(ii) Forget Term}}$$
$$+ \underbrace{\|\tilde{\mathbf{W}}\mathbf{K}_0 - \mathbf{M}_0\|^2}_{\text{(iii) Utility Term}}. \tag{4}$$

The zero term encourages the updated MLP outputs $\tilde{\mathbf{W}}\mathbf{K}_f$ to be as orthogonal as possible to $\mathbf{M}_f$, which encodes the original knowledge of the forget set. Please note that when $\|\mathbf{M}_f^{\top}(\tilde{\mathbf{W}}\mathbf{K}_f)\|^2 = 0$, the similarity between $\tilde{\mathbf{W}}\mathbf{K}_f$ and $\mathbf{M}_f$ is zero. The forget term aims to explicitly redirect the associative mapping of the forget set. By aligning the input keys $\mathbf{K}_f$ with a neutral target $\mathbf{M}_n$ (e.g., the representation of the "<EOS>" token), we actively guide the model to overwrite the undesired knowledge with a non-informative or terminal signal. This term ensures that the model does not merely suppress the original output but learns to map the sensitive inputs to a predefined "null" state, thereby effectively neutralizing the influence of the forget set.

Furthermore, the utility term $\|\tilde{\mathbf{W}}\mathbf{K}_0 - \mathbf{M}_0\|^2$ serves as a fidelity constraint to preserve the model's general capa-

---

[1] We utilize the `20220301.en` subset from `https://huggingface.co/datasets/wikimedia/wikipedia`.

bilities. It encourages the updated weight matrix $\tilde{\mathbf{W}}$ to maintain the original input-output associations for the remaining knowledge base $(\mathbf{K}_0, \mathbf{M}_0)$. By minimizing this term, we ensure that the unlearning process remains precise, modifying only the targeted factual associations while preventing catastrophic forgetting or degradation of the model's fundamental linguistic proficiency.

## 4.3. ZeroUnlearn: Null-Space Constrained Unlearning

To both simplify the objective of ZeroUnlearn and alleviate the trade-off issue, we introduce a new editing paradigm. In contrast to traditional methods that apply an additive perturbation to the original parameter matrix $\mathbf{W}$, we explore a multiplicative formulation by directly left-multiplying $\mathbf{W}$ with a projection matrix $\mathbf{D}$, namely $\tilde{\mathbf{W}} = \mathbf{DW}$. At this point, the original problem (Eq. 4) can be reformulated as

$$\mathbf{D}^* = \underset{\mathbf{D}}{\arg\min} \|\mathbf{M}_f^\top(\mathbf{DWK}_f)\|^2 + \|\mathbf{DWK}_f - \mathbf{M}_n\|^2$$
$$+ \|\mathbf{DWK}_0 - \mathbf{M}_0\|^2. \tag{5}$$

To ensure the zero term is identically zero, we aim to find an appropriate $\mathbf{D}$ in the right null space of $\mathbf{M}_f^\top$ such that $\mathbf{M}_f^\top \mathbf{D} = \mathbf{0}$. Specifically, we perform the singular value decomposition (SVD) of $\mathbf{M}_f^\top$, yielding $\mathbf{M}_f^\top = \mathbf{U\Sigma V}^\top$. Then we define the orthogonal projection matrix as $\mathbf{P} = \mathbf{I} - \mathbf{VV}^\top$. At this point, $\mathbf{P}$ lies in the right null space of $\mathbf{M}_f^\top$, i.e., $\mathbf{M}_f^\top \mathbf{P} = \mathbf{0}$. Therefore, by reparameterizing $\mathbf{D}$ as $\mathbf{D} = \mathbf{P}\tilde{\mathbf{D}}$, it follows that $\mathbf{D}$ also lies in the right null space of $\mathbf{M}_f^\top$. The updated optimization objective can be expressed as

$$\min_{\tilde{\mathbf{D}}} \|\mathbf{P}\tilde{\mathbf{D}}\mathbf{WK}_f - \mathbf{M}_n\|^2 + \|\mathbf{P}\tilde{\mathbf{D}}\mathbf{WK}_0 - \mathbf{M}_0\|^2. \tag{6}$$

In this manner, we elegantly avoid the trade-off between catastrophic forgetting and model capacity. Finally, in practical applications, we introduce an additional regularization term to ensure stable convergence of the model:

$$\tilde{\mathbf{D}}^* = \underset{\tilde{\mathbf{D}}}{\arg\min} \|\mathbf{P}\tilde{\mathbf{D}}\mathbf{WK}_f - \mathbf{M}_n\|^2 + \|\mathbf{P}\tilde{\mathbf{D}}\mathbf{WK}_0 - \mathbf{M}_0\|^2$$
$$+ \|\tilde{\mathbf{D}}\mathbf{W} - \mathbf{W}\|^2. \tag{7}$$

**Lemma 4.1** (Close-form solution for ZeroUnlearn). *The final optimization objective shown in Objective 7 admits a closed-form solution and can be expressed as follows:*

$$\tilde{\mathbf{D}}^* = \mathbf{P}(\mathbf{A} + \mathbf{W})\mathbf{W}^\top(\mathbf{W}(\mathbf{B} + \mathbf{I})\mathbf{W}^\top)^{-1}, \tag{8}$$

*where* $\mathbf{A} = \mathbf{M}_n\mathbf{K}_f^\top + \mathbf{M}_0\mathbf{K}_0^\top$ *and* $\mathbf{B} = \mathbf{K}_f\mathbf{K}_f^\top + \mathbf{K}_0\mathbf{K}_0^\top$. *Because* $\mathbf{P}$ *is an orthogonal projector satisfying* $\mathbf{P}^2 = \mathbf{P}$, *we have* $\mathbf{D}^* = \mathbf{P}\tilde{\mathbf{D}}^* = \tilde{\mathbf{D}}^*$.

This closed-form expression characterizes the optimal transformation $\tilde{\mathbf{D}}$ by balancing targeted knowledge erasure, utility preservation, and parameter stability through the following components. **(i) Target-Key Association Matrix (A).**

---

**Algorithm 1** ZeroUnlearn

1: **Input:** Utility set $\mathcal{E}_0 = \{\mathbf{t}_0^i\}$, Forget set $\mathcal{E}_f = \{(\mathbf{s}_f^i, \mathbf{r}_f^i, \mathbf{o}_f^i)\}$, Target state $\mathbf{M}_n$
2: **Output:** Modified generator without knowledge from $\mathcal{E}_f$
3: **for** each target layer to edit **do**
4:   *Phase 1: Knowledge Extraction*
5:   **for** $\mathbf{t}_0^i \in \mathcal{E}_0$ **do**
6:     $\mathbf{k}_0^i \leftarrow k(\mathbf{t}_0^i)$
7:   **end for**
8:   $\mathbf{K}_0 \leftarrow [\mathbf{k}_0^1, \ldots, \mathbf{k}_0^n]$
9:   **for** $\mathbf{s}_f^i \in \mathcal{E}_f$ **do**
10:    $\mathbf{k}_f^i \leftarrow \frac{1}{N_x}\sum_{j=1}^{N_x} k(\text{concat}[\mathbf{x}_j, \mathbf{s}_f^i])$
11:    /* $\mathbf{x}_j$ *is a random string prefix* */
12:   **end for**
13:   $\mathbf{K}_f \leftarrow [\mathbf{k}_f^1, \ldots, \mathbf{k}_f^n]$
14:   *Phase 2: Matrix Construction*
15:   $\mathbf{M}_0 \leftarrow \mathbf{WK}_0$
16:   $\mathbf{A} \leftarrow \mathbf{M}_n\mathbf{K}_f^\top + \mathbf{M}_0\mathbf{K}_0^\top$
17:    /* $\mathbf{M}_n$ *is the MLP output different from* $\mathbf{M}_f$ */
18:   $\mathbf{B} \leftarrow \mathbf{K}_f\mathbf{K}_f^\top + \mathbf{K}_0\mathbf{K}_0^\top$
19:   $\mathbf{M}_f^\top = \mathbf{U\Sigma V}^\top$
20:   $\mathbf{P} \leftarrow \mathbf{I} - \mathbf{VV}^\top$
21:   *Phase 3: Weight Update*
22:   $\mathbf{D}^* \leftarrow \mathbf{P}(\mathbf{A} + \mathbf{W})\mathbf{W}^\top(\mathbf{W}(\mathbf{B} + \mathbf{I})\mathbf{W}^\top)^{-1}$
23:   $\mathbf{W} \leftarrow \mathbf{D}^*\mathbf{W}$
24: **end for**

---

The matrix $\mathbf{A} = \mathbf{M}_n\mathbf{K}_f^\top + \mathbf{M}_0\mathbf{K}_0^\top$ represents the aggregated cross-correlation between the desired output targets and the input keys. The term $\mathbf{M}_n\mathbf{K}_f^\top$ encodes the redirection of forget-set inputs toward the nullifying state, while $\mathbf{M}_0\mathbf{K}_0^\top$ anchors the remaining knowledge to its original representations. **(ii) Key Second Moment (B).** The matrix $\mathbf{B} = \mathbf{K}_f\mathbf{K}_f^\top + \mathbf{K}_0\mathbf{K}_0^\top$ is the uncentered second moment matrix of the input keys. It captures the energy distribution and sample density within the key space. In the closed-form solution, the term $(\mathbf{W}(\mathbf{B} + \mathbf{I})\mathbf{W}^\top)^{-1}$ acts as a precision-weighted normalizer, ensuring that the weight update is appropriately scaled relative to the frequency and magnitude of the input features.

Thus, our unlearning paradigm is performed by left-multiplying $\mathbf{D}^*$ with the weight matrix $\mathbf{W}$ of the selected layer, without compromising model capacity. The strategy for locating the layers to be edited is described in the experimental section. The algorithmic procedure of ZeroUnlearn is presented in Algorithm 1. Here, $k(\cdot)$ denotes the function that extracts the final token of the subject to represent the key corresponding to a given piece of knowledge. In practice, we prepend randomly sampled prefixes to the subject in order to enhance generalization (Meng et al., 2022b).

# 5. Few-shot Constraints of ZeroUnlearn and an Alternative Solution

The efficacy of ZeroUnlearn in few-shot unlearning scenarios can be analyzed through the spectral properties and the rank of the projection matrix $\mathbf{P}$. Given that the forget set $\mathcal{E}_f$ contains a limited number of samples $n$, where $n \ll d$ (and $d$ is the hidden dimension of the model), the original knowledge matrix $\mathbf{M}_f \in \mathbb{R}^{d \times n}$ is inherently low-rank. Formally, let $r = \text{rank}(\mathbf{M}_f^\top) \le n$. The orthogonal projector $\mathbf{P} = \mathbf{I} - \mathbf{V}\mathbf{V}^\top$ is constructed from the $r$ dominant singular vectors of $\mathbf{M}_f^\top$. According to the rank-nullity theorem, the rank of the projection matrix is:

$$\text{rank}(\mathbf{P}) = d - r \ge d - n. \tag{9}$$

In the few-shot scenario, since $n$ is extremely small relative to $d$, $\text{rank}(\mathbf{P})$ remains near-maximal. This high dimensionality of the null space implies that the model retains $d - n$ degrees of freedom to perform the unlearning task. Geometrically, the "forbidden subspace" spanned by the forget set is a tiny, low-dimensional filament within the vast activation manifold. By constraining the update $\mathbf{D}$ to the null space of $\mathbf{M}_f^\top$, ZeroUnlearn ensures that the modification is accurate. This ensures that while the specific directions corresponding to $\mathbf{M}_f$ are neutralized (where the projection gain is zero), the vast majority of the weight matrix's expressive capacity remains untouched. Consequently, the model can overwrite sensitive knowledge with minimal impact on its fundamental linguistic proficiency, effectively resolving the trade-off between forgetting precision and general utility.

Meanwhile, to extend our framework to multi-sample unlearning scenarios, we propose an alternative scheme based on additive weight editing. By defining the updated weight matrix as $\tilde{\mathbf{W}} = \mathbf{W} + \mathbf{D}_m$, the optimization objective in Eq. 4, augmented with a regularization term, can be reformulated as:

$$\min_{\mathbf{D_m}} \|\mathbf{M}_f^\top((\mathbf{W} + \mathbf{D}_m)\mathbf{K}_f)\|^2 + \|(\mathbf{W} + \mathbf{D}_m)\mathbf{K}_f - \mathbf{M}_n\|^2$$
$$+ \|(\mathbf{W} + \mathbf{D}_m)\mathbf{K}_0 - \mathbf{M}_0\|^2 + \|\mathbf{D}_m\|^2, \tag{10}$$

where $\mathbf{D}_m$ represents the additive perturbation matrix. Similarly, to reconcile the trade-off between unlearning efficacy and general utility, we mandate that $\mathbf{K}_0$ resides in the right null space of the additive editing matrix $\mathbf{D}_m$. Specifically, we perform the SVD on the second moment $\mathbf{K}_0\mathbf{K}_0^\top$, yielding:

$$\mathbf{K}_0\mathbf{K}_0^\top = \mathbf{U}_m\mathbf{\Sigma}_m\mathbf{U}_m^\top. \tag{11}$$

We construct $\mathbf{U}_m'$ by extracting the eigenvectors from $\mathbf{U}_m$ that correspond to the zero eigenvalues. These vectors form an orthonormal basis for the null space, ensuring that any additive update $\mathbf{E}$ parameterized by $\mathbf{P}_m = \mathbf{U}_m'(\mathbf{U}_m')^\top$ satisfies the hard constraint $\mathbf{E}\mathbf{K}_0 = \mathbf{0}$. This projector $\mathbf{P}_m$ maps any vector onto the right null space of $\mathbf{K}_0^\top$. By reparameterizing the additive perturbation as $\mathbf{D}_m = \tilde{\mathbf{D}}_m\mathbf{P}_m$,

we ensure that: $\mathbf{D_m}\mathbf{K}_0 = \tilde{\mathbf{D}}_m\mathbf{P}_m\mathbf{K}_0 = \mathbf{0}$, which theoretically guarantees that the utility term in Eq. 10 vanishes identically. Consequently, the optimization problem for multi-sample unlearning is simplified to finding the optimal $\hat{\mathbf{E}}$ that minimizes the remaining forget-related terms:

$$\min_{\tilde{\mathbf{D}}_m} \|\mathbf{M}_f^\top((\mathbf{W} + \tilde{\mathbf{D}}_m\mathbf{P}_m)\mathbf{K}_f)\|^2$$
$$+ \|(\mathbf{W} + \tilde{\mathbf{D}}_m\mathbf{P}_m)\mathbf{K}_f - \mathbf{M}_n\|^2 + \|\tilde{\mathbf{D}}_m\mathbf{P}_m\|^2. \tag{12}$$

**Lemma 5.1** (Closed-form solution for Multiple Unlearning). *The optimization objective presented in Eq. 12 constitutes a Sylvester equation with respect to the effective update matrix. The optimal solution $\tilde{\mathbf{D}}_m^*$ admits a closed-form expression via the vectorization operator:*

$$vec(\tilde{\mathbf{D}}_m^*) = \left(\mathbf{H}^\top \otimes \mathbf{Q} + \mathbf{C}^\top \otimes \mathbf{I}\right)^{-1} vec(\mathbf{Z}), \tag{13}$$

*where $\otimes$ denotes the Kronecker product, and $\text{vec}(\cdot)$ is the vectorization operator. The matrices involved are defined as follows:*

$$\mathbf{Q} = \mathbf{M}_f\mathbf{M}_f^\top + \mathbf{I}, \quad \mathbf{H} = \mathbf{P}_m\mathbf{K}_f\mathbf{K}_f^\top\mathbf{P}_m^\top, \quad \mathbf{C} = \mathbf{P}_m\mathbf{P}_m^\top,$$
$$\mathbf{Z} = \mathbf{M}_n\mathbf{K}_f^\top\mathbf{P}_m^\top - \mathbf{Q}\mathbf{W}\mathbf{K}_f\mathbf{K}_f^\top\mathbf{P}_m^\top. \tag{14}$$

After computing the vector solution, $\tilde{\mathbf{D}}_m^*$ is recovered by reshaping the result to the original matrix dimensions.

## 5.1. Complexity Analysis and Practical Optimization

While Lemma 5.1 provides a theoretically rigorous global optimum for the multi-sample unlearning objective, directly computing the closed-form solution is computationally prohibitive for modern LLMs.

**Computational Bottleneck.** The primary bottleneck lies in the inversion of the term $\mathbf{K}_{\text{kron}} = \mathbf{H}^\top \otimes \mathbf{Q} + \mathbf{C}^\top \otimes \mathbf{I}$. Let $d$ denote the hidden dimension of the model. The Kronecker product results in a matrix $\mathbf{K}_{\text{kron}} \in \mathbb{R}^{d^2 \times d^2}$. Standard matrix inversion algorithms scale cubically with the matrix dimension. Therefore, the time complexity for solving the vectorized equation is:

$$\mathcal{O}((d^2)^3) = \mathcal{O}(d^6). \tag{15}$$

Furthermore, the space complexity required to store $\mathbf{K}_{\text{kron}}$ is $\mathcal{O}(d^4)$. For a typical LLM where $d > 1000$, storing this matrix would require huge memory, rendering the closed-form solution intractable.

**Gradient-Based Approximation.** To circumvent these limitations, we adopt an iterative optimization strategy. Since the objective function in Eq. 12 is convex with respect to $\tilde{\mathbf{D}}_m$ (composed of quadratic terms), Gradient Descent (GD) is guaranteed to converge to the global optimum. By employing GD, we avoid the explicit construction of the Kronecker product. The gradients can be computed efficiently using standard backpropagation, with a computational complexity of $\mathcal{O}(d^2)$ per iteration. We refer to this multi-sample unlearning approach as ZeroUnlearn-GD.

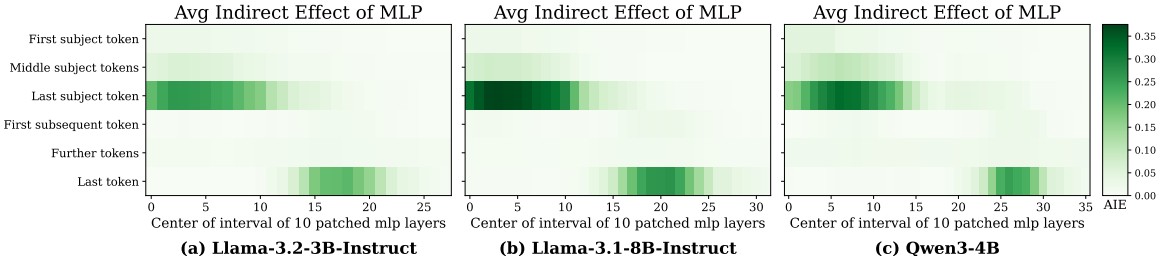

*Figure 2.* Causal tracing for knowledge localization.

# 6. Experiments

## 6.1. Settings

**Base Model and Baselines.** We employ three widely adopted models, Llama-3.2-3B-Instruct (Llama-3.2), Llama-3.1-8B-Instruct (Llama-3.1) (Grattafiori et al., 2024) and Qwen-3-4B (Qwen-3) (Yang et al., 2025), as our base models. Since knowledge editing-based approaches typically utilize only the forget set, we adopt GA (Jang et al., 2023), which adheres to the same data constraint. Regarding editing-based methods, we evaluate four representative baselines: FT (Zhu et al., 2020), ROME (Meng et al., 2022a), MEMIT (Meng et al., 2022b), and AlphaEdit (Fang et al., 2024). For a comprehensive description of these baselines, please refer to Appendix B.

**Datasets and Metrics.** To validate the effectiveness of our method, we utilize the relation-pair dataset MCF (Meng et al., 2022a), alongside two question answering datasets: ZsRE (Levy et al., 2017) and MQUAKE (Zhong et al., 2024). To quantitatively assess the unlearning capabilities, we adopt the following metrics: **(i) Efficacy (Eff.):** This metric evaluates the residual retention of the specific knowledge intended to be forgotten. It is defined as the average probability that the unlearned model $f_{\theta'}$ generates the original target answer $y_f$ given the input query $x_f$ from the forget set $\mathcal{D}_f$. Formally, this can be expressed as:

$$\text{Eff} = \frac{1}{|\mathcal{D}_f|} \sum_{(x_f, y_f) \in \mathcal{D}_f} P_{\theta'}(y_f \mid x_f), \qquad (16)$$

where $P_{\theta'}(y_f \mid x_f)$ denotes the likelihood assigned by the model to the ground-truth label. A lower score ($\downarrow$) indicates better unlearning performance, implying that the model effectively ceases to produce the original sensitive response. **(ii) Generalization (Gen.):** This measures the consistency of unlearning across paraphrased queries. It assesses whether the model successfully suppresses the sensitive information even when the input is rephrased. **(iii) Specificity (Spe.):** This metric evaluates the preservation of non-targeted knowledge. We measure the model's accuracy on the neighborhood query, where a higher score ($\uparrow$) indicates that the unlearning is precise and does not cause collateral damage to related knowledge. **(iv) PPL:** To ensure

the model's general linguistic capabilities are not degraded, we compute the perplexity (PPL) on a hold-out corpus (e.g., WikiText). A lower PPL ($\downarrow$) signifies that the model maintains its generative quality. The detailed information about datasets are provided in C. Moreover, all results for Qwen-3, all results on the MQUAKE dataset, and the performance of Llama-3.1 in the multiple unlearning scenario are provided in Appendix F.

## 6.2. Target Layer Identification

To achieve precise unlearning, identifying the location of knowledge within the pre-trained weights is a prerequisite. Following the paradigm of model interpretability (Meng et al., 2022a), we employ Causal Tracing to analyze the contribution of different model components. Using 1000 prompts in (Meng et al., 2022a), we calculate the Average Indirect Effect (AIE) for the MLP module at each layer. The process involves two steps: (i) corrupting the subject representation in the input to degrade the model's prediction probability for the target fact; and (ii) restoring the activation of specific MLP layers to their original state during inference. The degree to which the probability of the correct answer recovers quantifies the layer's causal importance. As illustrated in Figure 2, the results across different models exhibit a consistent localized pattern. We observe that the significant causal effects are not uniformly distributed but are concentrated within some continuous layers. Based on this empirical evidence, we designate these high-impact layers as the target scope for our unlearning intervention (Appendix E). This selection aims to modify the parameters most responsible for factual retrieval while minimizing perturbations to unrelated components.

## 6.3. Results of Few-Shot Unlearning

In the few-shot scenario, we conduct experiments using ten random seeds (ranging from 1 to 10), where 50 samples are randomly selected per seed for the unlearning process. Table 1 presents the comprehensive performance of our proposed ZeroUnlearn against various baselines.

**Unlearning Efficacy.** As indicated by the efficacy (Eff.) metric, ZeroUnlearn achieves state-of-the-art performance

*Table 1.* Few-shot unlearning results of ZeroUnlearn on MCF and ZsRE datasets.

| Method | Model | MCF | | | | ZsRE | | | |
|---|---|---|---|---|---|---|---|---|---|
| | | Eff.↓ | Gen.↓ | Spe.↑ | PPL↓ | Eff.↓ | Gen.↓ | Spe.↑ | PPL↓ |
| Base | | 18.20±3.84 | 20.30±5.33 | 19.60±3.47 | 12.88±0.00 | 32.82±4.09 | 32.23±4.16 | 28.12±2.65 | 12.88±0.00 |
| GA | Llama-3.2 | 2.00±3.34 | 1.80±2.89 | 1.06±1.79 | >1000 | 1.41±1.36 | 1.16±1.42 | 3.53±1.41 | >1000 |
| FT | | 0.00±0.00 | 0.00±0.00 | 0.00±0.00 | 18.25±1.28 | 28.83±3.96 | 27.70±3.34 | 26.80±2.57 | 13.24±0.11 |
| ROME | | 18.20±3.84 | 20.30±5.37 | 19.50±3.51 | 12.88±0.20 | 32.80±4.20 | 32.17±4.09 | 28.05±2.66 | 12.89±0.20 |
| MEMIT | | 17.00±4.22 | 18.30±4.92 | 19.20±3.62 | 12.86±0.02 | 32.32±4.00 | 31.17±4.61 | 28.01±2.60 | 12.89±0.02 |
| AlphaEdit | | 2.60±2.37 | 11.80±3.94 | 18.36±3.63 | 12.84±0.02 | 29.59±3.95 | 29.90±4.67 | 27.80±2.77 | 12.88±0.04 |
| **ZeroUnlearn** | | 0.40±0.80 | 4.60±2.24 | 14.90±2.93 | 13.06±0.18 | 27.85±3.87 | 27.52±3.87 | 27.73±2.70 | 13.08±0.06 |
| Base | | 24.60±5.29 | 22.80±4.35 | 21.96±4.28 | 7.47±0.00 | 40.42±4.92 | 36.84±4.24 | 29.87±2.30 | 7.47±0.00 |
| GA | Llama-3.1 | 1.20±1.83 | 0.90±1.81 | 0.26±0.72 | >1000 | 0.27±0.61 | 0.27±0.61 | 0.00±0.00 | >1000 |
| FT | | 0.00±0.00 | 0.00±0.00 | 0.00±0.00 | 10.23±0.67 | 31.36±2.19 | 30.91±2.96 | 26.99±2.01 | 8.16±0.08 |
| ROME | | 24.40±5.04 | 22.60±4.10 | 21.86±4.28 | 7.48±0.01 | 40.46±4.85 | 36.84±4.16 | 29.99±2.37 | 7.48±0.01 |
| MEMIT | | 9.60±4.63 | 16.20±4.07 | 21.08±4.24 | 7.51±0.03 | 35.15±3.99 | 34.60±3.15 | 30.05±2.46 | 7.48±0.03 |
| AlphaEdit | | 0.20±0.60 | 7.80±2.27 | 19.74±4.20 | 7.49±0.05 | 34.12±4.16 | 34.19±3.33 | 29.93±2.49 | 7.48±0.07 |
| **ZeroUnlearn** | | 0.00±0.00 | 4.60±2.11 | 16.82±3.64 | 7.77±0.06 | 32.67±3.43 | 32.39±3.34 | 29.67±2.36 | 7.76±0.10 |

*Table 2.* Multiple unlearning results of ZeroUnlearn-GD on MCF and ZsRE datasets.

| Method | Model | MCF | | | | ZsRE | | | |
|---|---|---|---|---|---|---|---|---|---|
| | | Eff.↓ | Gen.↓ | Spe.↑ | PPL↓ | Eff.↓ | Gen.↓ | Spe.↑ | PPL↓ |
| Base | | 22.10 | 19.60 | 20.59 | 12.88 | 33.76 | 33.44 | 29.55 | 12.88 |
| GA | Llama-3.2 | 0.00 | 0.00 | 0.00 | >1000 | 0.00 | 0.00 | 0.11 | >1000 |
| FT | | 0.00 | 0.00 | 0.00 | 63.70 | 25.13 | 24.52 | 24.62 | 15.57 |
| ROME | | 21.90 | 19.55 | 20.47 | 12.89 | 33.67 | 33.06 | 29.59 | 12.90 |
| MEMIT | | 13.80 | 14.45 | 17.75 | 12.77 | 29.78 | 29.60 | 29.16 | 13.01 |
| AlphaEdit | | 1.40 | 10.20 | 15.68 | 12.78 | 27.66 | 27.54 | 28.04 | 13.25 |
| **ZeroUnlearn-GD** | | 0.00 | 5.10 | 12.41 | 13.05 | 25.29 | 25.22 | 25.32 | 13.60 |

in erasing target knowledge. On the MCF dataset with Llama-3.1, our method reduces the efficacy to **0%**, demonstrating complete removal of the sensitive information. In contrast, traditional knowledge editing methods like ROME and MEMIT struggle to effectively unlearn in this setting (e.g., ROME retains 24.40% efficacy, similar to the base model). While AlphaEdit shows improvement over other editors, ZeroUnlearn consistently outperforms it, achieving significantly lower residual knowledge retention.

**Preserving Model Capabilities.** A critical challenge in unlearning is avoiding catastrophic forgetting. Naive approaches like GA achieve low efficacy scores but at the cost of destroying the model's linguistic abilities, as evidenced by the exploded PPL (>1000) and collapsed specificity (Spe.). Similarly, FT suffers from overfitting, resulting in a complete loss of specificity (0% on MCF) and degraded perplexity. ZeroUnlearn, conversely, maintains a PPL score comparable to the base model and retains high specificity. This confirms that our method performs surgical unlearning without causing collateral damage to the model's general generative capabilities or unrelated knowledge.

**Generalization-Specificity Trade-off.** Ideally, unlearning

should be surgical, removing only the target without affecting neighborhood knowledge. As shown in Table 1, methods that fail to effectively unlearn (e.g., ROME and MEMIT) naturally retain high Specificity, similar to the base model. However, among methods that achieve significant unlearning efficacy (Eff. ≈ 0), ZeroUnlearn demonstrates a superior capability to preserve unrelated knowledge. While we observe a moderate decrease in specificity compared to the base model, our method avoids the catastrophic collapse seen in GA and FT (where Spe. drops to near zero). This indicates that ZeroUnlearn strikes a more practical trade-off, successfully erasing sensitive information while maintaining a reasonable level of neighborhood knowledge.

### 6.4. Results of Multiple Unlearning

In the multiple unlearning scenario, we select 1,000 samples to perform the unlearning process. Table 2 presents the performance of ZeroUnlearn-GD in the challenging multi-sample unlearning scenario. As evidenced by the results on MCF, our gradient-based variant demonstrates exceptional efficacy and scalability, achieving **0%** Eff. on Llama-3.2. This indicates a complete erasure of targeted batch knowl-

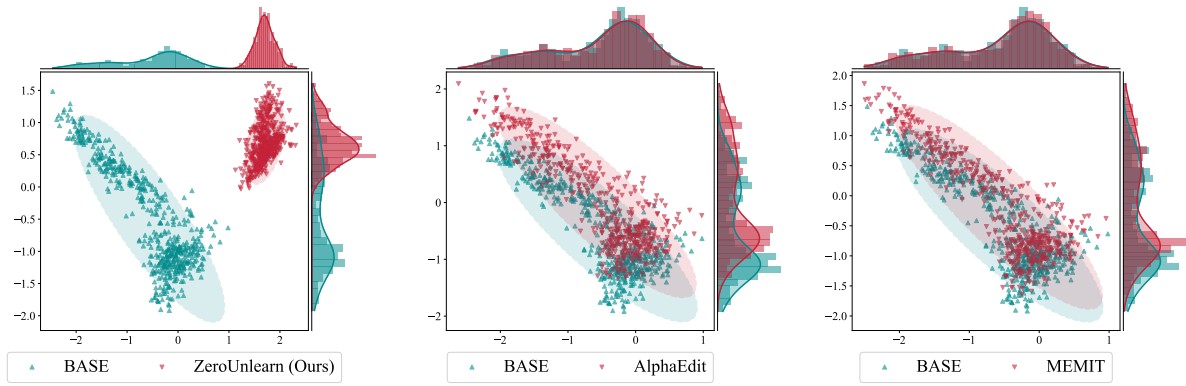

*Figure 3.* PCA visualization of MLP representation shifts at Layer 16 of Llama-3.2 on the MCF dataset.

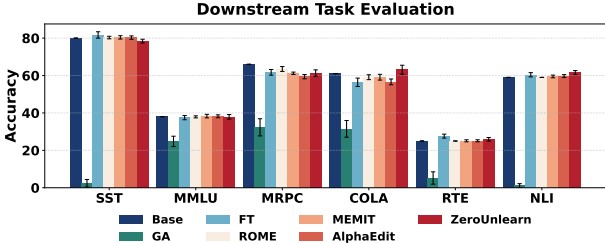

*Figure 4.* Evaluation of general capabilities on Llama-3.2.

edge, significantly outperforming dedicated mass-editing baselines like MEMIT and AlphaEdit, which struggle to eliminate residual information. Crucially, ZeroUnlearn-GD achieves this thorough unlearning without the catastrophic model collapse observed in optimization-based approaches; while GA and FT lead to exploded perplexity and a total loss of specificity, our method maintains the model's linguistic capabilities within a stable, functional range.

Regarding locality, ZeroUnlearn-GD experiences a notable reduction in Specificity, which is an expected trade-off for achieving perfect erasure in large batches. However, unlike optimization-based baselines (GA and FT) that suffer from total locality collapse (Specificity $\approx 0$), our method maintains a functional level of neighborhood knowledge, effectively balancing the aggressive removal of sensitive data with the preservation of general model utility.

### 6.5. Representation Visualization

To visually verify the unlearning effect, we project the MLP output of a certain layer on the forget set samples into a 2D space using PCA (Maćkiewicz & Ratajczak, 1993). As shown in Figure 3, the results reveal a distinct contrast. For ZeroUnlearn (left), the representations of the unlearned samples (red) are clearly separated from the original base model's representations (cyan), forming an independent cluster distinct from the original distribution. This con-

firms that our method effectively removes the sensitive information from the feature space. Conversely, for baselines like AlphaEdit and MEMIT, the unlearned representations heavily overlap with the original ones, indicating that these methods fail to fundamentally alter the internal encoding of the targeted knowledge. More Visualization results are provided in Appendix G.

### 6.6. Downstream Evaluation

To verify that ZeroUnlearn preserves the model's general capabilities, we evaluate performance across six diverse downstream tasks (SST (Socher et al., 2013b), MMLU (Hendrycks et al., 2020),MRPC (Dolan & Brockett, 2005), COLA (Warstadt et al., 2019), RTE (Bentivogli et al., 2009) and NLI (Williams et al., 2018)). As shown in Figure 4, GA suffers from catastrophic forgetting, exhibiting a near-total collapse in accuracy on tasks like SST and NLI. In contrast, ZeroUnlearn maintains performance levels statistically comparable to the original base model across all benchmarks. This confirms that our method achieves surgical unlearning without degrading the model's fundamental reasoning and linguistic competencies.

### 6.7. Practical Efficiency of ZeroUnlearn

We measure the practical runtime and memory usage of the few-shot closed-form update across different forget-set sizes. All experiments were conducted on Llama-3.2. The results show that the SVD step itself is very lightweight: even when the forget-set size increases from 10 to 1000, the average SVD time remains below 0.3 seconds on MCF/ZsRE and below 0.6 seconds on MQUAKE, while the corresponding memory usage only increases modestly from about 13.8 GB to 14.1 GB. We also report the end-to-end cost of the full editing procedure. As expected, runtime grows approximately linearly with forget-set size, from about 0.04 h at 10 samples to 3.35–3.82 h at 1000 samples across datasets. Total memory remains stable at roughly 14.9–17.4 GB. These re-

sults suggest that the closed-form update is not the practical bottleneck; the main cost comes from key/value extraction and layer-wise editing.

# 7. Conclusion

In this work, we introduced ZeroUnlearn, a novel methodology that redefines machine unlearning as a precise knowledge remapping process. By leveraging a multiplicative parameter update mechanism, ZeroUnlearn projects sensitive representations into an orthogonal null space, thereby ensuring effective erasure while minimizing collateral damage to the model's general utility. We further derived a closed-form solution for efficient few-shot updates and extended it to ZeroUnlearn-GD for batch processing. Extensive empirical evaluations across multiple LLMs demonstrate that our approach significantly outperforms existing baselines, achieving a superior balance between unlearning and utility.

# Impact Statement

This paper introduces a novel framework for precise knowledge erasure in large language models. Our work has the potential to significantly improve AI safety by enabling the removal of toxic content, hallucinations, and private data without retraining. ZeroUnlearn contributes to the development of more trustworthy and legally compliant AI systems. We do not foresee immediate negative societal consequences that must be specifically highlighted here.

# Acknowledgements

The project was supported by National Key&D Program of China (No. 2022ZD0160501), Natural Science Foundation of Fujian Province of China (No. 2024J011001), and the Public Technology Service Platform Project of Xiamen (No.3502Z20231043). We also thank the reviewers for their insightful comments.

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

# A. Notation

Table 3. Summary of symbols used throughout the paper. Vectors and matrices are in bold.

| Symbol | Meaning |
|---|---|
| $\mathcal{D}_f = \{(x_i, y_i)\}_{i=1}^n$ | Forget set (samples whose influence should be removed). |
| $f_\theta, \theta \in \Theta$ | Pre-trained language model parameterized by $\theta$. |
| $\mathcal{U}(\cdot)$ | Unlearning operator; $\theta' = \mathcal{U}(\theta, \mathcal{D}_f)$. |
| $\theta', \Delta\theta$ | Updated parameters and parameter change ($\Delta\theta = \theta' - \theta$). |
| $L$ | Number of transformer layers. |
| $l \in \{1, \ldots, L\}$ | Layer index. |
| $x$ | Input token (or token sequence). |
| $\mathbf{h}^{l-1}, \mathbf{h}^l$ | Hidden state at layers $l-1$ and $l$. |
| $\mathbf{a}^l$ | Causal self-attention output at layer $l$. |
| $\mathbf{m}^l$ (or $\mathbf{m}$) | MLP output at layer $l$ (often omitting layer superscript). |
| $\mathrm{Norm}(\cdot), \sigma(\cdot)$ | Layer normalization and activation function. |
| $\mathbf{W}_{\mathrm{up}}, \mathbf{W}_{\mathrm{down}}$ | MLP expansion/projection matrices; $\mathbf{W}_{\mathrm{down}}$ maps MLP features to $\mathbf{m}$. |
| $\mathbf{W}$ | Shorthand for the edited weight matrix (typically $\mathbf{W}_{\mathrm{down}}$ at a chosen layer). |
| $\mathbf{k}$ | MLP feature vector (pre-$\mathbf{W}$), e.g., $\mathbf{k} = \sigma(\mathbf{W}_{\mathrm{in}}\mathrm{Norm}(\cdot))$ so that $\mathbf{m} = \mathbf{W}\mathbf{k}$. |
| $(s, r, o)$ | Knowledge triple: subject, relation, object. |
| $k(\cdot)$ | Function extracting the key vector for a knowledge instance (e.g., last-token feature of the subject). |
| $\mathcal{E}_f = \{(s_i^f, r_i^f, o_i^f)\}$ | Forget knowledge set in triple form. |
| $\mathcal{E}_0 = \{t_i^0\}$ | Utility/anchor set used to preserve general behavior. |
| $\mathbf{K}_f, \mathbf{M}_f$ | Stacked forget keys/outputs with columns $\{\mathbf{k}_i^f\}$ and $\{\mathbf{m}_i^f\}$. |
| $\mathbf{K}_0, \mathbf{M}_0$ | Stacked utility keys/outputs with columns $\{\mathbf{k}_i^0\}$ and $\{\mathbf{m}_i^0\}$. |
| $\mathbf{M}_n$ | Target (nullifying) state/representation (e.g., an <EOS> terminal state). |
| $\tilde{\mathbf{W}}$ | Updated weight matrix after editing. |
| $\|\cdot\|_2$ | Euclidean / Frobenius norm (context-dependent). |
| $\mathbf{D}$ | Multiplicative left-update matrix with $\tilde{\mathbf{W}} = \mathbf{D}\mathbf{W}$. |
| $\mathbf{P}$ | Orthogonal projector onto the right null space of $\mathbf{M}_f^\top$ (to enforce orthogonality). |
| $\mathbf{M}_f^\top = \mathbf{U}\boldsymbol{\Sigma}\mathbf{V}^\top$ | SVD of $\mathbf{M}_f^\top$; $\mathbf{U}, \boldsymbol{\Sigma}, \mathbf{V}$ are SVD factors. |
| $\mathbf{P} = \mathbf{I} - \mathbf{V}\mathbf{V}^\top$ | Null-space projector constructed from $\mathbf{V}$. |
| $\tilde{\mathbf{D}}$ | Re-parameterized update with $\mathbf{D} = \mathbf{P}\tilde{\mathbf{D}}$. |
| $r$ | Rank of $\mathbf{M}_f^\top$ (typically $r \leq n$). |
| $d$ | Hidden/MLP feature dimension. |
| $\mathbf{D}_m$ | Additive update matrix for multi-sample variant, $\tilde{\mathbf{W}} = \mathbf{W} + \mathbf{D}_m$. |
| $\mathbf{P}_m$ | Projector onto the right null space of $\mathbf{K}_0^\top$ (to satisfy $\mathbf{D}_m\mathbf{K}_0 = \mathbf{0}$). |
| $\mathrm{vec}(\cdot), \otimes$ | Vectorization operator and Kronecker product. |
| $\mathbf{I}$ | Identity matrix (dimension implied by context). |
| $\mathbf{0}$ | All-zeros vector or matrix (dimension implied by context). |

# B. Baselines Details

In this section, we provide detailed descriptions of the baseline methods employed in our comparative evaluation. These baselines encompass both optimization-based unlearning approaches and state-of-the-art knowledge editing techniques.

**GA (Gradient Ascent) (Jang et al., 2023).** GA is a standard baseline for machine unlearning that operates by reversing the standard training objective. Instead of minimizing the prediction error, GA maximizes the loss function on the forget set $\mathcal{D}_f$. While effective at erasing specific data traces, this unconstrained maximization often leads to the destruction of the model's language modeling capabilities and catastrophic forgetting of unrelated knowledge.

**FT (Fine-Tuning) (Zhu et al., 2020).** In the context of knowledge editing, FT serves as a naive baseline where the model parameters are updated via standard gradient descent to map the specific input $\mathbf{k}$ to a new target output $\mathbf{v}$ (e.g., the target token or an empty response).

**ROME (Rank-One Model Editing) (Meng et al., 2022a).** ROME treats the feed-forward networks in transformer models as key-value associative memories. It first utilizes causal tracing to locate the specific layer and neuron responsible for a factual association. Subsequently, it computes a rank-one update to the FFN weights. This update is explicitly designed to force the modified layer to map the subject representation to the desired target vector, while simultaneously minimizing

interference with other memories stored in the model.

**MEMIT (Mass-Editing Memory in a Transformer)** (Meng et al., 2022b). MEMIT extends the principles of ROME to the multi-edit setting. It distributes the knowledge update across multiple layers of the Transformer to increase capacity. Mathematically, MEMIT formulates the batch editing problem as a least-squares optimization with an equality constraint for the new memories. It aggregates the update directions from thousands of samples and applies a closed-form solution to inject large batches of knowledge simultaneously.

**AlphaEdit** (Fang et al., 2024). AlphaEdit is a recently proposed improvement over ROME and MEMIT that addresses the "over-correction" issue in projection-based editing. It introduces a null-space constraint mechanisms during the covariance statistics accumulation phase. By strictly constraining the update direction to be orthogonal to the preserved knowledge subspace, AlphaEdit achieves higher specificity and stability, effectively minimizing the side effects on neighborhood knowledge compared to standard least-squares approaches.

## C. Dataset Details

To comprehensively evaluate the performance of our proposed method, we utilize two distinct categories of datasets. The first category focuses on assessing the efficacy of knowledge unlearning. The second category consists of standard benchmarks to evaluate the model's general utility, ensuring that the unlearning process does not compromise the model's fundamental reasoning and linguistic capabilities.

### C.1. Benchmarks for Knowledge Unlearning

We select three widely used datasets to construct the forget set ($\mathcal{D}_f$). For each dataset, we partition the data to evaluate the trade-off between forgetting specific facts and retaining relevant knowledge.

- **MCF (Multi-CounterFact)** (Meng et al., 2022a): A large-scale dataset designed to evaluate counterfactual knowledge editing. It contains diverse factual statements that allow us to test the model's ability to update or erase specific associations.

- **ZsRE (Zero-Shot Relation Extraction)** (Levy et al., 2017): This dataset is derived from Question-Answering tasks and is a standard benchmark for measuring model editing performance. It was originally proposed for relation extraction (Ma et al., 2026). It requires the model to answer questions based on specific relations, providing a robust test for precise unlearning.

- **MQUAKE** (Zhong et al., 2024): Although originally designed for assessing multi-hop knowledge editing, we adapt this dataset to a single-hop setting analogous to ZsRE. Specifically, we decompose the multi-hop reasoning chains into atomic, single-hop question-answer pairs. This processing strategy allows us to strictly evaluate the unlearning efficacy on diverse factual relations without the confounding factors of multi-hop reasoning, serving as a robust complement to ZsRE for testing direct fact erasure.

### C.2. Benchmarks for General Utility

To ensure that our method maintains the general capabilities of the LLM, we evaluate the model on six diverse tasks covering sentiment analysis, semantic matching, and logical reasoning. These include MMLU and selected tasks from the GLUE benchmark (Wang et al., 2019).

- **MMLU (Massive Multi-task Language Understanding)** (Hendrycks et al., 2021): A comprehensive evaluation suite designed to measure multi-task accuracy. It assesses the model's broad knowledge and reasoning ability under zero-shot and few-shot settings across various domains.

- **SST (The Stanford Sentiment Treebank)** (Socher et al., 2013a): A single-sentence classification task involving movie reviews. It evaluates the model's ability to identify and classify sentiment labels correctly.

- **MRPC (Microsoft Research Paraphrase Corpus)** (Dolan & Brockett, 2005): A well-known benchmark for text matching and semantic similarity assessment. The objective is to determine whether a given pair of sentences is semantically equivalent.

- **RTE (Recognizing Textual Entailment)** (Bentivogli et al., 2009): This task involves natural language inference, requiring the model to determine if a premise sentence logically entails a hypothesis sentence.

- **COLA (Corpus of Linguistic Acceptability)** (Warstadt et al., 2019): A single-sentence classification task where sentences are annotated as either grammatically acceptable or unacceptable, testing the model's linguistic competence.

- **NLI (Natural Language Inference)** (Williams et al., 2018): This task focuses on natural language understanding, requiring the model to infer the logical relationship (entailment, contradiction, or neutral) between pairs of sentences.

# D. Proof of Lemmas

In this section, we provide detailed mathematical derivations for the closed-form solutions presented in Lemma 4.1 and Lemma 5.1. We utilize standard matrix calculus notation, where the Frobenius norm is defined as $\|\mathbf{X}\|_F^2 = \mathrm{Tr}(\mathbf{X}^\top \mathbf{X})$.

### D.1. Proof of Lemma 4.1 (ZeroUnlearn)

*Proof.* Recall the optimization objective for ZeroUnlearn. We seek the optimal transformation $\tilde{\mathbf{D}}$ that minimizes the reconstruction errors in the projected subspace defined by $\mathbf{P}$, with regularization on the weight changes:

$$\mathcal{L}(\tilde{\mathbf{D}}) = \left\|\mathbf{P}\tilde{\mathbf{D}}\mathbf{W}\mathbf{K}_f - \mathbf{M}_n\right\|_F^2 + \left\|\mathbf{P}\tilde{\mathbf{D}}\mathbf{W}\mathbf{K}_0 - \mathbf{M}_0\right\|_F^2 + \left\|\tilde{\mathbf{D}}\mathbf{W} - \mathbf{W}\right\|_F^2. \tag{17}$$

Since $\mathbf{P}$ projects onto the null space of $\mathbf{M}_f^\top$, the "Zero Term" $\left\|\mathbf{M}_f^\top(\mathbf{P}\tilde{\mathbf{D}}\mathbf{W}\mathbf{K}_f)\right\|^2$ vanishes by design ($\mathbf{M}_f^\top\mathbf{P} = \mathbf{0}$) and is omitted from the derivative calculation for brevity. We assume $\mathbf{P}$ is an orthogonal projection matrix, satisfying $\mathbf{P}^\top = \mathbf{P}$ and $\mathbf{P}^2 = \mathbf{P}$.

We compute the gradient of $\mathcal{L}$ with respect to $\tilde{\mathbf{D}}$:

$$\begin{aligned}
\frac{1}{2}\nabla_{\tilde{\mathbf{D}}}\mathcal{L} = \ &\mathbf{P}^\top(\mathbf{P}\tilde{\mathbf{D}}\mathbf{W}\mathbf{K}_f - \mathbf{M}_n)(\mathbf{W}\mathbf{K}_f)^\top \\
&+ \mathbf{P}^\top(\mathbf{P}\tilde{\mathbf{D}}\mathbf{W}\mathbf{K}_0 - \mathbf{M}_0)(\mathbf{W}\mathbf{K}_0)^\top \\
&+ (\tilde{\mathbf{D}}\mathbf{W} - \mathbf{W})\mathbf{W}^\top.
\end{aligned} \tag{18}$$

Using the property $\mathbf{P}^\top = \mathbf{P}$ and $\mathbf{P}^\top\mathbf{P} = \mathbf{P}$, the terms simplify. The stationarity condition $\nabla_{\tilde{\mathbf{D}}}\mathcal{L} = \mathbf{0}$ becomes:

$$\mathbf{P}(\tilde{\mathbf{D}}\mathbf{W}\mathbf{K}_f - \mathbf{M}_n)(\mathbf{W}\mathbf{K}_f)^\top + \mathbf{P}(\tilde{\mathbf{D}}\mathbf{W}\mathbf{K}_0 - \mathbf{M}_0)(\mathbf{W}\mathbf{K}_0)^\top + (\tilde{\mathbf{D}}\mathbf{W} - \mathbf{W})\mathbf{W}^\top = \mathbf{0}. \tag{19}$$

To ensure the solution $\tilde{\mathbf{D}}$ lies within the valid subspace (Range of $\mathbf{P}$) and satisfies the optimality condition for the constrained problem (Projected Gradient = 0), we left-multiply the entire equation by $\mathbf{P}$. This projects the regularization gradient onto the valid subspace:

$$\mathbf{P}^2(\dots) + \mathbf{P}^2(\dots) + \mathbf{P}(\tilde{\mathbf{D}}\mathbf{W} - \mathbf{W})\mathbf{W}^\top = \mathbf{0}. \tag{20}$$

Since $\mathbf{P}^2 = \mathbf{P}$, we can factor $\mathbf{P}$ out of the entire expression:

$$\mathbf{P}\left[(\tilde{\mathbf{D}}\mathbf{W}\mathbf{K}_f - \mathbf{M}_n)(\mathbf{W}\mathbf{K}_f)^\top + (\tilde{\mathbf{D}}\mathbf{W}\mathbf{K}_0 - \mathbf{M}_0)(\mathbf{W}\mathbf{K}_0)^\top + (\tilde{\mathbf{D}}\mathbf{W} - \mathbf{W})\mathbf{W}^\top\right] = \mathbf{0}. \tag{21}$$

We construct the solution by first solving for the unconstrained optimizer of the expression inside the brackets, and then projecting it. Setting the term inside the brackets to zero:

$$\tilde{\mathbf{D}}\mathbf{W}(\mathbf{K}_f\mathbf{K}_f^\top + \mathbf{K}_0\mathbf{K}_0^\top + \mathbf{I})\mathbf{W}^\top = (\mathbf{M}_n\mathbf{K}_f^\top + \mathbf{M}_0\mathbf{K}_0^\top + \mathbf{W})\mathbf{W}^\top. \tag{22}$$

Let $\mathbf{A} = \mathbf{M}_n\mathbf{K}_f^\top + \mathbf{M}_0\mathbf{K}_0^\top$ and $\mathbf{B} = \mathbf{K}_f\mathbf{K}_f^\top + \mathbf{K}_0\mathbf{K}_0^\top$. The equation simplifies to:

$$\tilde{\mathbf{D}}\left(\mathbf{W}(\mathbf{B} + \mathbf{I})\mathbf{W}^\top\right) = (\mathbf{A} + \mathbf{W})\mathbf{W}^\top. \tag{23}$$

Solving for $\tilde{\mathbf{D}}$ yields a base solution. To satisfy the constraint $\tilde{\mathbf{D}} \in \mathrm{Range}(\mathbf{P})$, we apply the projection $\mathbf{P}$ to this base solution. It can be verified that $\tilde{\mathbf{D}}^* = \mathbf{P}\tilde{\mathbf{D}}$ satisfies the original projected gradient equation (Eq. 21):

$$\tilde{\mathbf{D}}^* = \mathbf{P}(\mathbf{A} + \mathbf{W})\mathbf{W}^\top\left(\mathbf{W}(\mathbf{B} + \mathbf{I})\mathbf{W}^\top\right)^{-1}. \tag{24}$$

$\square$

### D.2. Proof of Lemma 5.1

*Proof.* We start with the optimization objective for multi-sample unlearning (Eq. 12). Since $\mathbf{P}_m\mathbf{K}_0 = \mathbf{0}$, the utility term vanishes. The optimization objective is given by:

$$\mathcal{L}(\tilde{\mathbf{D}}_m) = \left\|\mathbf{M}_f^\top(\mathbf{W} + \tilde{\mathbf{D}}_m\mathbf{P}_m)\mathbf{K}_f\right\|_F^2 + \left\|(\mathbf{W} + \tilde{\mathbf{D}}_m\mathbf{P}_m)\mathbf{K}_f - \mathbf{M}_n\right\|_F^2 + \left\|\tilde{\mathbf{D}}_m\mathbf{P}_m\right\|_F^2. \tag{25}$$

Since the objective function is convex with respect to $\tilde{\mathbf{D}}_m$, we derive the optimal solution by setting the gradient $\nabla_{\tilde{\mathbf{D}}_m}\mathcal{L}$ to zero. We utilize the matrix derivative identity $\frac{\partial\|\mathbf{X}\mathbf{Y}+\mathbf{Z}\|_F^2}{\partial\mathbf{X}} = 2(\mathbf{X}\mathbf{Y} + \mathbf{Z})\mathbf{Y}^\top$ and $\frac{\partial\|\mathbf{A}\mathbf{X}\mathbf{B}\|_F^2}{\partial\mathbf{X}} = 2\mathbf{A}^\top\mathbf{A}\mathbf{X}\mathbf{B}\mathbf{B}^\top$. Differentiating $\mathcal{L}(\tilde{\mathbf{D}}_m)$ with respect to $\tilde{\mathbf{D}}_m$ yields:

$$\begin{aligned}
\frac{\partial\mathcal{L}}{\partial\tilde{\mathbf{D}}_m} = {}& 2\mathbf{M}_f\mathbf{M}_f^\top(\mathbf{W} + \tilde{\mathbf{D}}_m\mathbf{P}_m)\mathbf{K}_f\mathbf{K}_f^\top\mathbf{P}_m^\top \\
& + 2(\mathbf{W} + \tilde{\mathbf{D}}_m\mathbf{P}_m)\mathbf{K}_f\mathbf{K}_f^\top\mathbf{P}_m^\top - 2\mathbf{M}_n\mathbf{K}_f^\top\mathbf{P}_m^\top \\
& + 2\tilde{\mathbf{D}}_m\mathbf{P}_m\mathbf{P}_m^\top.
\end{aligned} \tag{26}$$

Setting the gradient to zero and dividing by 2, we rearrange the terms to isolate $\tilde{\mathbf{D}}_m$. We group the terms involving $\mathbf{W}$ and move them to the right-hand side, while keeping terms with $\tilde{\mathbf{D}}_m$ on the left:

$$(\mathbf{M}_f\mathbf{M}_f^\top + \mathbf{I})\tilde{\mathbf{D}}_m\mathbf{P}_m\mathbf{K}_f\mathbf{K}_f^\top\mathbf{P}_m^\top + \tilde{\mathbf{D}}_m\mathbf{P}_m\mathbf{P}_m^\top = \mathbf{M}_n\mathbf{K}_f^\top\mathbf{P}_m^\top - (\mathbf{M}_f\mathbf{M}_f^\top + \mathbf{I})\mathbf{W}\mathbf{K}_f\mathbf{K}_f^\top\mathbf{P}_m^\top. \tag{27}$$

To simplify the notation, we define the following auxiliary matrices $\mathbf{Q}, \mathbf{H}, \mathbf{C}$, and the constant matrix $\mathbf{Z}$:

$$\mathbf{Q} = \mathbf{M}_f\mathbf{M}_f^\top + \mathbf{I}, \tag{28}$$

$$\mathbf{H} = \mathbf{P}_m\mathbf{K}_f\mathbf{K}_f^\top\mathbf{P}_m^\top, \tag{29}$$

$$\mathbf{C} = \mathbf{P}_m\mathbf{P}_m^\top, \tag{30}$$

$$\mathbf{Z} = \mathbf{M}_n\mathbf{K}_f^\top\mathbf{P}_m^\top - \mathbf{Q}\mathbf{W}\mathbf{K}_f\mathbf{K}_f^\top\mathbf{P}_m^\top. \tag{31}$$

Substituting these into the gradient equation, the optimization condition reduces to a generalized Sylvester equation:

$$\mathbf{Q}\tilde{\mathbf{D}}_m\mathbf{H} + \tilde{\mathbf{D}}_m\mathbf{C} = \mathbf{Z}. \tag{32}$$

To solve for $\tilde{\mathbf{D}}_m$, we apply the vectorization operator $\text{vec}(\cdot)$ and use the property of the Kronecker product $\text{vec}(\mathbf{A}\mathbf{X}\mathbf{B}) = (\mathbf{B}^\top \otimes \mathbf{A})\text{vec}(\mathbf{X})$. This transforms the matrix equation into the following linear system:

$$(\mathbf{H}^\top \otimes \mathbf{Q})\text{vec}(\tilde{\mathbf{D}}_m) + (\mathbf{C}^\top \otimes \mathbf{I})\text{vec}(\tilde{\mathbf{D}}_m) = \text{vec}(\mathbf{Z}). \tag{33}$$

Merging the terms acting on $\text{vec}(\tilde{\mathbf{D}}_m)$, we obtain the closed-form solution:

$$\text{vec}(\tilde{\mathbf{D}}_m^*) = \left(\mathbf{H}^\top \otimes \mathbf{Q} + \mathbf{C}^\top \otimes \mathbf{I}\right)^{-1}\text{vec}(\mathbf{Z}). \tag{34}$$

$\square$

# E. Details for Layer Identification

Figure 5 illustrates the variation in the average indirect effect (AIE) for each token across all layers. We observe that for MLP outputs, the layers where the last subject token exhibits the peak AIE are often concentrated in the model's early (bottom) layers. However, our experiments reveal that editing these lower layers significantly compromises the model's general capabilities. In practice, specifically for Llama-3.1 and Llama-3.2, where the peak AIE of the last subject token is located in the bottom layers, we target the layers where the last token achieves its maximum AIE, as these are concentrated in the middle section of the model.

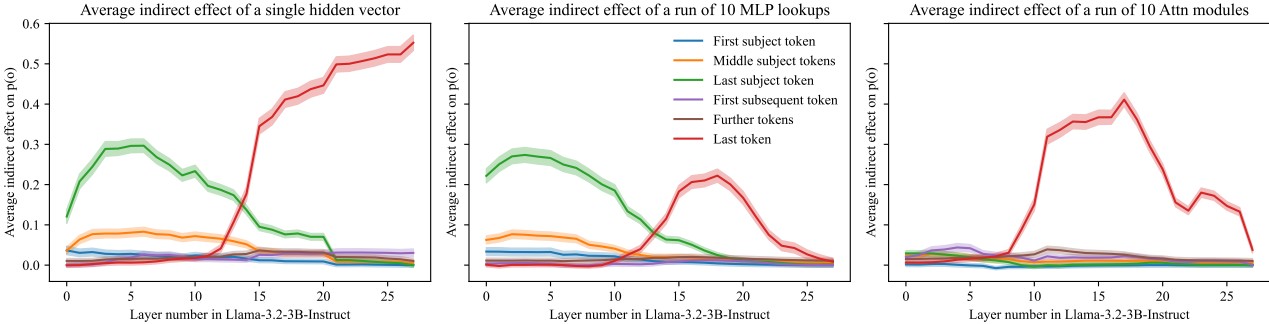

*Figure 5.* **Component-wise causal tracing analysis on Llama-3.2.** The line plots visualize the Average Indirect Effect (AIE) of single hidden vectors (left), MLP lookups (middle), and Attention modules (right) across layers. The results highlight distinct roles: MLP layers in the early stages (layers 0-10) show high causal influence on the *last subject token* (green line), suggesting knowledge retrieval, whereas the *last token* (red line) dominates the causal effect in later layers, particularly in attention modules.

To further dissect the internal information flow across different architectures, we visualize the Average Indirect Effect of Attention modules and hidden states in Figure 6 and Figure 7, respectively.

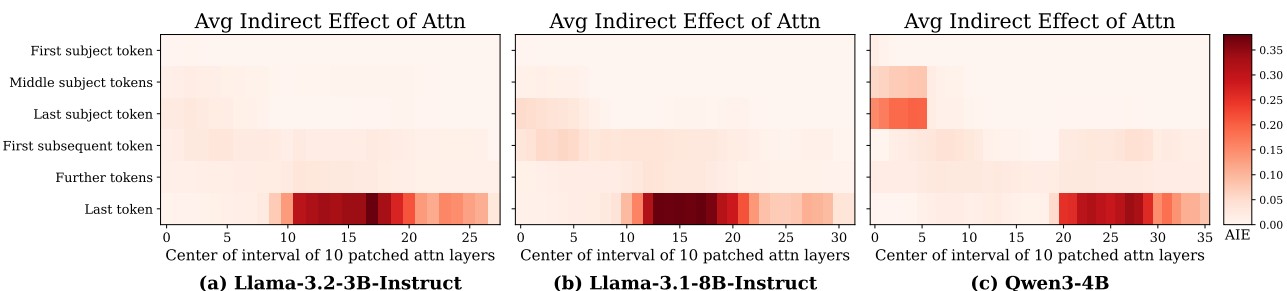

*Figure 6.* Average Indirect Effect of Attention modules across different architectures.

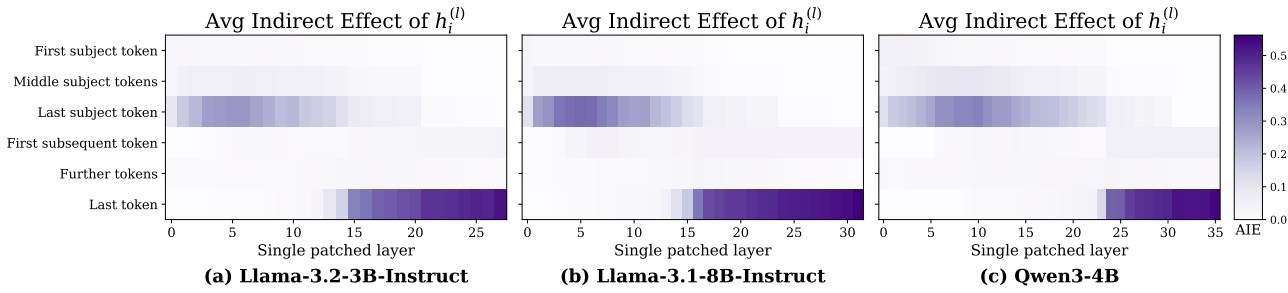

*Figure 7.* Layer-wise causal efficacy of hidden states ($h_i^{(l)}$).

## E.1. Ablation Study

To evaluate the contribution of the core components in ZeroUnlearn, we conduct an ablation study focusing on the role of the neutral target state $\mathbf{M}_n$. We compare the full ZeroUnlearn method against a variant denoted as "w/o $\mathbf{M}_n$", where the optimization relies solely on the null-space projection constraint without explicitly guiding the output towards a neutral value. The results on the ZsRE dataset across Llama-3.2, Llama-3.1, and Qwen-3 are summarized in Table 4.

**Impact of the Neutral Target State ($\mathbf{M}_n$).**    As shown in Table 4, incorporating $\mathbf{M}_n$ yields a consistent improvement in unlearning performance across all model architectures. The inclusion of the target state significantly reduces the Efficacy (Eff.) and Generalization (Gen.) scores (where lower values indicate better unlearning). For instance, on Llama-3.1, the Efficacy score improves from 36.02 to 32.67. This suggests that merely projecting the weights into the null space of the sensitive knowledge is insufficient for complete erasure. The $\mathbf{M}_n$ term acts as a directional guide, actively steering the model's behavior towards a neutral state , thereby ensuring a more thorough removal of the targeted associative mapping.

*Table 4.* Ablation results of ZeroUnlearn on ZsRE dataset.

| Method | Model | ZsRE | | | |
|---|---|---|---|---|---|
| | | Eff.↓ | Gen.↓ | Spe.↑ | PPL↓ |
| **ZeroUnlearn** | Llama-3.2 | $27.85_{\pm 3.87}$ | $31.25_{\pm 0.49}$ | $27.73_{\pm 2.70}$ | $13.08_{\pm 0.06}$ |
| w/o $\mathbf{M}_n$ | | $30.03_{\pm 3.59}$ | $32.30_{\pm 0.65}$ | $27.44_{\pm 2.18}$ | $13.04_{\pm 0.10}$ |
| **ZeroUnlearn** | Llama-3.1 | $32.67_{\pm 3.43}$ | $37.09_{\pm 1.07}$ | $29.67_{\pm 2.36}$ | $7.76_{\pm 0.10}$ |
| w/o $\mathbf{M}_n$ | | $36.02_{\pm 4.23}$ | $39.16_{\pm 0.86}$ | $29.41_{\pm 2.48}$ | $7.76_{\pm 0.05}$ |
| **ZeroUnlearn** | Qwen-3 | $25.35_{\pm 2.80}$ | $29.22_{\pm 0.68}$ | $27.28_{\pm 3.13}$ | $10.96_{\pm 0.24}$ |
| w/o $\mathbf{M}_n$ | | $25.64_{\pm 3.30}$ | $29.10_{\pm 0.63}$ | $27.00_{\pm 3.39}$ | $10.88_{\pm 0.19}$ |

# F. Complete Results

## F.1. Few-shot unlearning results of Qwen-3

*Table 5.* Few-shot unlearning results of ZeroUnlearn on MCF dataset.

| Method | Model | MCF | | | |
|---|---|---|---|---|---|
| | | Eff.↓ | Gen.↓ | Spe.↑ | PPL↓ |
| Base | | $14.40_{\pm 4.45}$ | $14.10_{\pm 3.18}$ | $12.84_{\pm 2.21}$ | 11.21 |
| GA | Qwen-3 | $0.80_{\pm 1.83}$ | $0.50_{\pm 1.50}$ | $0.58_{\pm 1.74}$ | >1000 |
| FT | | $0.00_{\pm 0.00}$ | $2.90_{\pm 1.97}$ | $6.54_{\pm 2.21}$ | $12.49_{\pm 0.70}$ |
| ROME | | $14.20_{\pm 3.94}$ | $13.90_{\pm 3.11}$ | $13.10_{\pm 2.25}$ | $11.31_{\pm 0.20}$ |
| MEMIT | | $0.80_{\pm 1.33}$ | $3.80_{\pm 2.04}$ | $12.82_{\pm 2.09}$ | $11.23_{\pm 0.06}$ |
| AlphaEdit | | $0.60_{\pm 0.91}$ | $2.60_{\pm 1.74}$ | $12.52_{\pm 2.01}$ | $11.24_{\pm 0.14}$ |
| **ZeroUnlearn** | | $0.00_{\pm 0.00}$ | $2.20_{\pm 1.60}$ | $9.38_{\pm 1.69}$ | $11.19_{\pm 0.26}$ |

*Table 6.* Few-shot unlearning results of ZeroUnlearn on ZsRE dataset.

| Method | Model | ZsRE | | | |
|---|---|---|---|---|---|
| | | Eff.↓ | Gen.↓ | Spe.↑ | PPL↓ |
| Base | | $30.63_{\pm 2.97}$ | $31.13_{\pm 3.49}$ | $27.69_{\pm 3.27}$ | 11.21 |
| GA | Qwen-3 | $3.17_{\pm 2.05}$ | $2.81_{\pm 1.59}$ | $7.10_{\pm 1.76}$ | >1000 |
| FT | | $28.89_{\pm 4.47}$ | $27.80_{\pm 3.80}$ | $27.98_{\pm 3.26}$ | $11.97_{\pm 0.40}$ |
| ROME | | $30.66_{\pm 3.20}$ | $30.93_{\pm 3.23}$ | $27.70_{\pm 3.37}$ | $11.25_{\pm 0.50}$ |
| MEMIT | | $28.97_{\pm 3.16}$ | $29.55_{\pm 3.01}$ | $27.83_{\pm 3.21}$ | $11.20_{\pm 0.06}$ |
| AlphaEdit | | $28.52_{\pm 3.30}$ | $28.29_{\pm 3.60}$ | $27.82_{\pm 3.14}$ | $11.20_{\pm 0.13}$ |
| **ZeroUnlearn** | | $25.35_{\pm 2.80}$ | $25.53_{\pm 3.17}$ | $27.28_{\pm 3.13}$ | $10.96_{\pm 0.24}$ |

*Table 7.* Few-shot unlearning results of ZeroUnlearn on MQUAKE dataset.

| Method | Model | MQUAKE | |
|---|---|---|---|
| | | Eff.↓ | PPL↓ |
| Base | | $29.16_{\pm3.60}$ | 11.21 |
| GA | Qwen-3 | $1.54_{\pm0.42}$ | $455.19_{\pm412.34}$ |
| FT | | $24.91_{\pm4.47}$ | $11.55_{\pm0.40}$ |
| ROME | | $28.92_{\pm3.59}$ | $11.28_{\pm0.78}$ |
| MEMIT | | $25.55_{\pm4.03}$ | $11.27_{\pm0.06}$ |
| AlphaEdit | | $24.62_{\pm3.58}$ | $11.29_{\pm0.13}$ |
| **ZeroUnlearn** | | $22.98_{\pm3.64}$ | $11.29_{\pm0.26}$ |

## F.2. Multiple unlearning results of Llama-3.2 on MQUAKE

*Table 8.* Multiple unlearning results of ZeroUnlearn on MQUAKE dataset.

| Method | Model | MQUAKE | |
|---|---|---|---|
| | | Eff.↓ | PPL↓ |
| Base | | 48.33 | 12.88 |
| GA | Llama3-2 | 1.70 | >1000 |
| FT | | 15.56 | 14.36 |
| ROME | | 49.21 | 12.82 |
| MEMIT | | 26.66 | 12.94 |
| AlphaEdit | | 24.71 | 13.04 |
| **ZeroUnlearn-GD** | | 24.55 | 12.87 |

## F.3. Multiple unlearning results of Llama-3.1

*Table 9.* Multiple unlearning results of Llama-3.1.

| Method | Model | MCF | | | | ZsRE | | | | MQuAKE | |
|---|---|---|---|---|---|---|---|---|---|---|---|
| | | Eff.↓ | Gen.↓ | Spe.↑ | PPL↓ | Eff.↓ | Gen.↓ | Spe.↑ | PPL↓ | Eff.↓ | PPL↓ |
| Base | | 27.70 | 22.50 | 23.35 | 7.48 | 40.71 | 39.12 | 31.61 | 7.48 | 64.94 | 7.48 |
| GA | Llama-3.1 | 0.00 | 0.00 | 0.00 | >1000 | 0.10 | 0.11 | 1.08 | >1000 | 0.03 | >1000 |
| FT | | 0.00 | 0.00 | 0.00 | 275.30 | 20.42 | 19.70 | 21.45 | 9.92 | 14.44 | 9.02 |
| ROME | | 27.30 | 22.70 | 23.17 | 7.48 | 40.77 | 38.93 | 31.72 | 7.48 | 62.59 | 7.46 |
| MEMIT | | 5.30 | 5.55 | 19.51 | 7.70 | 34.60 | 34.24 | 31.49 | 7.63 | 29.91 | 7.35 |
| AlphaEdit | | 0.50 | 2.85 | 15.46 | 8.01 | 33.53 | 33.09 | 30.58 | 7.87 | 29.71 | 7.24 |
| **ZeroUnlearn-GD** | | 0.00 | 2.30 | 13.31 | 7.94 | 31.44 | 31.50 | 29.60 | 8.20 | 27.49 | 8.24 |

## F.4. Multiple unlearning results of Qwen-3

*Table 10.* Multiple unlearning results of Qwen-3.

| Method | Model | MCF | | | | ZsRE | | | | MQuAKE | |
|---|---|---|---|---|---|---|---|---|---|---|---|
| | | Eff.↓ | Gen.↑ | Spe.↑ | PPL↓ | Eff.↓ | Gen.↑ | Spe.↑ | PPL↓ | Eff.↓ | PPL↓ |
| Base | | 14.90 | 12.85 | 13.30 | 11.21 | 31.44 | 31.08 | 28.98 | 11.21 | 26.75 | 11.21 |
| GA | Qwen-3 | 0.00 | 0.00 | 0.00 | >1000 | 0.09 | 0.08 | 0.53 | >1000 | 0.34 | >1000 |
| FT | | 0.00 | 0.00 | 0.00 | 714.52 | 24.68 | 23.90 | 27.04 | 12.48 | 18.08 | 12.27 |
| ROME | | 14.60 | 13.15 | 13.41 | 11.38 | 31.50 | 31.25 | 29.14 | 11.25 | 26.70 | 11.16 |
| MEMIT | | 0.70 | 4.50 | 11.35 | 11.54 | 30.33 | 30.07 | 28.62 | 11.29 | 23.14 | 11.63 |
| AlphaEdit | | 0.40 | 3.30 | 11.04 | 13.03 | 28.55 | 28.89 | 28.07 | 10.90 | 23.21 | 12.94 |
| **ZeroUnlearn-GD** | | 0.40 | 3.00 | 9.52 | 11.63 | 28.04 | 27.73 | 27.26 | 11.71 | 23.37 | 11.57 |

# G. Complete PCA Visualization

For completeness, we provide the full PCA visualizations of the MLP representation shifts at the critical editing layers for all evaluated models. Figure 8, Figure 9, and Figure 10 illustrate the distinct geometric changes in Llama-3.1, Llama-3.2, and Qwen-3, respectively.

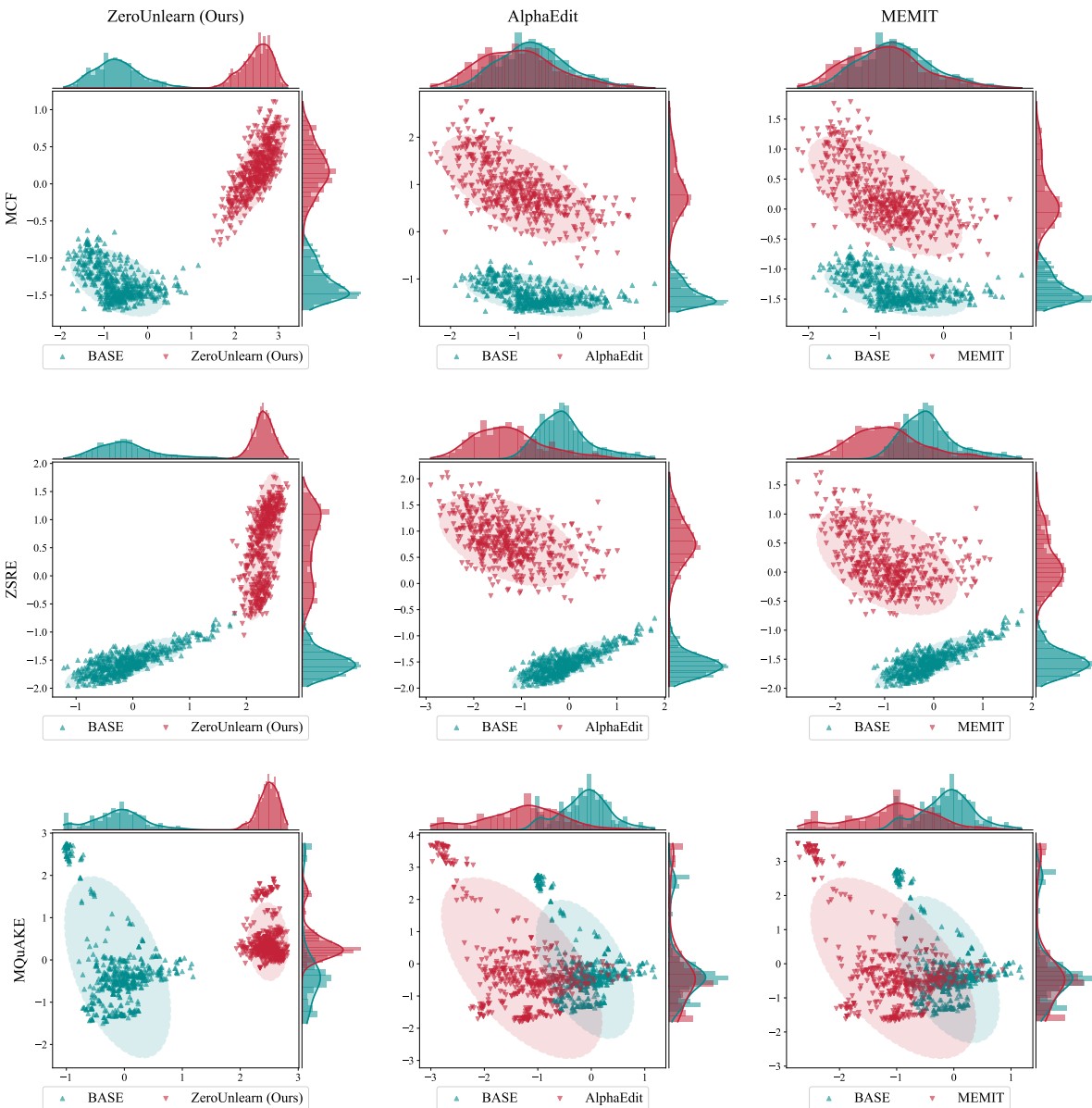

*Figure 8.* PCA visualization of MLP representation shifts at Layer 19 of Llama-3.1.

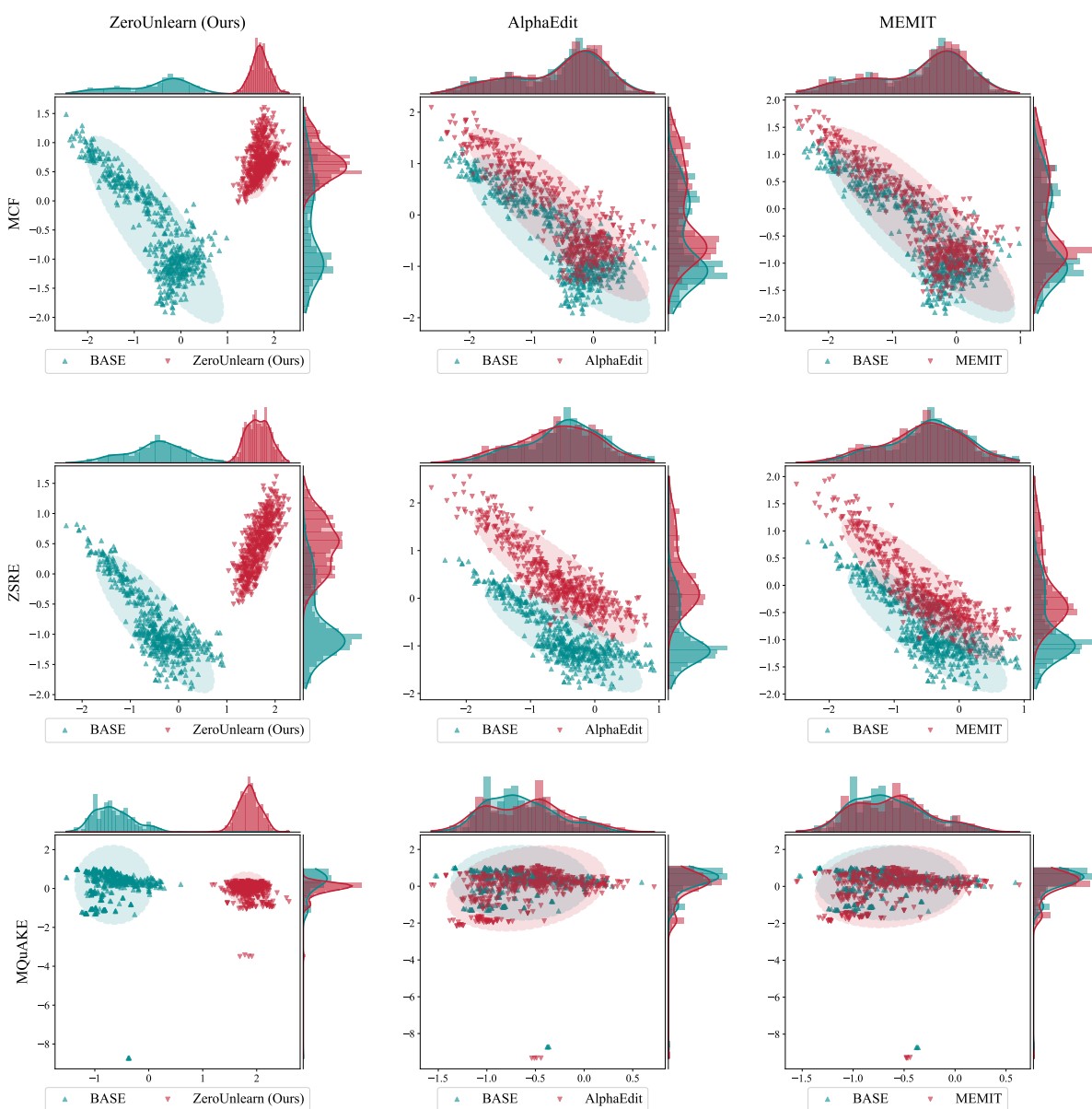

*Figure 9.* PCA visualization of MLP representation shifts at Layer 16 of Llama-3.2.

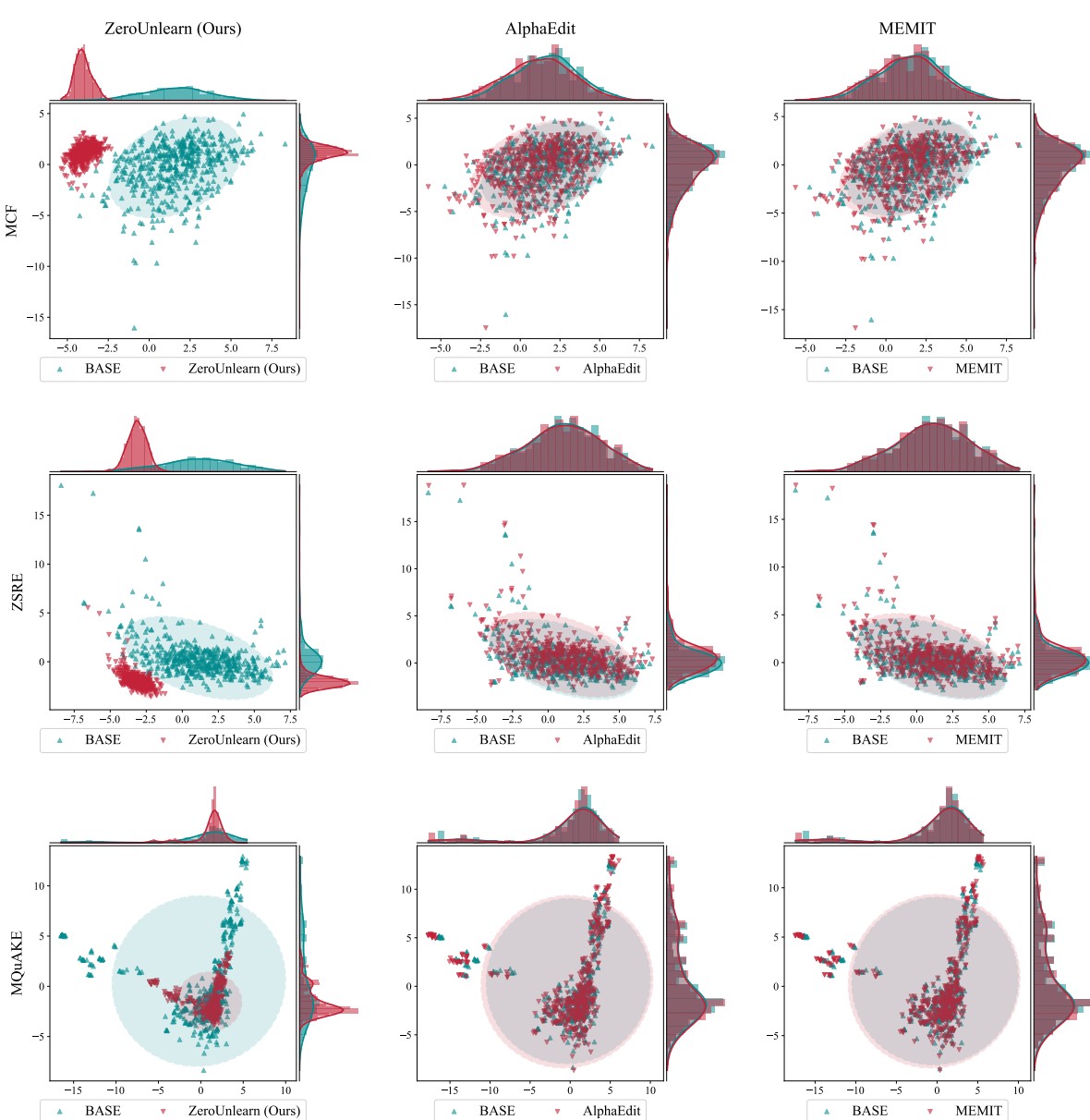

*Figure 10.* PCA visualization of MLP representation shifts at Layer 9 of Qwen3-4B.

# H. Additional Experiments

This appendix section incorporates the additional experiments conducted during the rebuttal stage. Specifically, we include evaluations on RWKU (Cao et al., 2024), comparisons with NPO variants (Zhang et al., 2024), sample-size sensitivity analysis and additional exploration of the neutral target state $\mathbf{M}_n$. All supplementary experiments were conducted on Llama-3.2. Since ZeroUnlearn and baselines only updates the down-projection matrices of three FFN layers, we implement two NPO variants for fairness: (i) NPO, which updates the same three layers, and (ii) NPO-full, which updates all parameters.

## H.1. Evaluation on RWKU

This section includes additional experiments on the RWKU benchmark. We report unlearning performance across three difficulty levels and compare the proposed method with GA, FT, ROME, MEMIT, AlphaEdit, NPO, and NPO-full.

*Table 11.* Unlearning results on RWKU (Level 1).

| Method | Eff.↓ | PPL↓ |
|---|---|---|
| Base | $39.66_{\pm0.17}$ | 12.87 |
| GA | $\mathbf{1.62}_{\pm17.1}$ | $976.00_{\pm6.3}$ |
| FT | $41.35_{\pm0.95}$ | $13.44_{\pm0.14}$ |
| ROME | $38.72_{\pm2.83}$ | $12.90_{\pm0.08}$ |
| MEMIT | $37.99_{\pm1.22}$ | $12.86_{\pm0.02}$ |
| AlphaEdit | $37.17_{\pm2.12}$ | $12.81_{\pm0.04}$ |
| NPO | $39.07_{\pm0.59}$ | $12.93_{\pm0.01}$ |
| NPO-full | $35.91_{\pm1.48}$ | $13.22_{\pm0.14}$ |
| **ZeroUnlearn** | $\underline{35.70}_{\pm3.79}$ | $13.18_{\pm0.18}$ |

*Table 12.* Unlearning results on RWKU (Level 2).

| Method | Eff.↓ | PPL↓ |
|---|---|---|
| Base | $39.12_{\pm0.17}$ | 12.87 |
| GA | $\mathbf{1.56}_{\pm2.8}$ | >1000 |
| FT | $38.09_{\pm1.02}$ | $12.92_{\pm0.07}$ |
| ROME | $38.51_{\pm1.77}$ | $12.91_{\pm0.09}$ |
| MEMIT | $38.01_{\pm1.42}$ | $12.87_{\pm1.80}$ |
| AlphaEdit | $35.53_{\pm2.82}$ | $12.86_{\pm0.06}$ |
| NPO | $38.98_{\pm0.27}$ | $12.89_{\pm0.01}$ |
| NPO-full | $35.56_{\pm2.51}$ | $13.00_{\pm0.09}$ |
| **ZeroUnlearn** | $\underline{35.47}_{\pm2.40}$ | $13.10_{\pm0.14}$ |

*Table 13.* Unlearning results on RWKU (Level 3).

| Method | Eff.↓ | PPL↓ |
|---|---|---|
| Base | $39.12_{\pm0.17}$ | 12.87 |
| GA | $\mathbf{2.64}_{\pm4.53}$ | > 1000 |
| FT | $40.62_{\pm1.56}$ | $13.01_{\pm0.06}$ |
| ROME | $39.07_{\pm2.88}$ | $12.92_{\pm0.09}$ |
| MEMIT | $38.93_{\pm1.99}$ | $12.86_{\pm1.30}$ |
| AlphaEdit | $37.45_{\pm2.62}$ | $12.84_{\pm4.10}$ |
| NPO | $39.71_{\pm0.43}$ | $12.91_{\pm0.01}$ |
| NPO-full | $37.66_{\pm2.36}$ | $13.05_{\pm0.08}$ |
| **ZeroUnlearn** | $\underline{34.01}_{\pm4.75}$ | $13.10_{\pm0.18}$ |

## H.2. Performance of NPO on Knowledge Editing Datasets

This section includes the experiments comparing NPO variants on the knowledge editing datasets. NPO fine-tunes the same three layers as our method, while NPO-full fine-tunes all layers.

*Table 14.* Unlearning results of NPO on the `MCF` dataset.

| Method | Eff.↓ | Gen.↓ | Spe.↑ | PPL↓ |
|---|---|---|---|---|
| NPO | 17.00±4.40 | 18.30±4.73 | 17.90±3.30 | 12.92±0.12 |
| NPO-full | 16.44±4.69 | 18.22±5.88 | **18.40**±4.05 | 13.03±0.11 |
| **ZeroUnlearn** | **0.40**±0.80 | **4.60**±2.24 | 14.90±2.93 | 13.06±0.18 |

*Table 15.* Unlearning results of NPO on the `ZsRE` dataset

| Method | Eff.↓ | Gen.↓ | Spe.↑ | PPL↓ |
|---|---|---|---|---|
| NPO | 30.77±3.72 | 30.15±4.24 | **28.09**±2.47 | 12.91±0.96 |
| NPO-full | 30.44±4.09 | 29.90±3.88 | 27.71±2.67 | 13.03±0.07 |
| **ZeroUnlearn** | **27.85**±3.87 | **27.52**±3.87 | 27.73±2.70 | 13.08±0.06 |

*Table 16.* Unlearning results of NPO on the `MQUAKE` dataset

| Method | Eff.↓ | PPL↓ |
|---|---|---|
| NPO | 32.33±5.23 | 12.9±0.01 |
| NPO-full | 30.64±4.63 | 12.93±0.07 |
| **ZeroUnlearn** | **24.51**±3.37 | 13.15±0.16 |

## H.3. Sample Size Sensitivity of ZeroUnlearn

We vary the number of forgotten samples and report the corresponding unlearning and utility metrics on the evaluated datasets.

*Table 17.* Unlearning results on the `MCF` dataset under different sample sizes.

| # Forgotten Samples | Eff.↓ | Gen.↓ | Spe.↑ | PPL↓ |
|---|---|---|---|---|
| 10 | 1.00±3.00 | 7.00±5.09 | 20.60±4.86 | 13.10±0.11 |
| 25 | 0.80±1.60 | 5.00±2.86 | 17.52±3.71 | 13.06±0.13 |
| 50 | 0.40±0.80 | 4.60±2.24 | 14.90±2.93 | 13.06±0.18 |
| 100 | 0.50±0.81 | 5.15±1.78 | 14.72±2.31 | 13.05±0.19 |
| 200 | 0.90±0.70 | 5.90±0.77 | 13.29±0.86 | 13.25±0.15 |
| 500 | 3.25±0.43 | 6.22±0.25 | 12.53±0.78 | 13.77±0.38 |
| 1000 | 6.73±0.42 | 8.45±0.66 | 12.20±0.49 | 14.25±0.41 |

*Table 18.* Unlearning results on the `ZsRE` dataset under different sample sizes.

| # Forgotten Samples | Eff.↓ | Gen.↓ | Spe.↑ | PPL↓ |
|---|---|---|---|---|
| 10 | 24.82±13.60 | 25.67±13.69 | 28.51±7.47 | 13.07±0.07 |
| 25 | 24.97±5.39 | 25.60±5.92 | 27.25±4.79 | 13.13±0.84 |
| 50 | 27.85±3.87 | 27.52±3.87 | 27.73±2.70 | 13.08±0.06 |
| 100 | 26.74±2.13 | 27.17±1.77 | 27.78±2.73 | 13.15±0.10 |
| 200 | 26.53±1.82 | 26.54±2.01 | 26.98±1.73 | 13.18±0.12 |
| 500 | 24.23±0.11 | 23.76±0.28 | 24.39±1.01 | 13.69±0.35 |
| 1000 | 20.67±0.34 | 19.85±0.25 | 20.48±0.44 | 14.50±0.55 |

*Table 19.* Unlearning results on the ZsRE dataset under different sample sizes.

| # Forgotten Samples | Eff.↓ | PPL↓ |
|---|---|---|
| 10 | $24.98_{\pm 9.91}$ | $13.05_{\pm 0.06}$ |
| 25 | $24.67_{\pm 6.72}$ | $13.03_{\pm 0.09}$ |
| 50 | $24.51_{\pm 3.37}$ | $13.15_{\pm 0.16}$ |
| 100 | $23.01_{\pm 2.53}$ | $13.18_{\pm 0.20}$ |
| 200 | $21.75_{\pm 2.28}$ | $13.27_{\pm 0.27}$ |
| 500 | $21.56_{\pm 1.66}$ | $13.31_{\pm 0.19}$ |
| 1000 | $19.28_{\pm 1.34}$ | $13.61_{\pm 0.22}$ |

## H.4. Further Exploration of Neutral Target $M_n$

This section includes additional experiments on the choice of the neutral target state $M_n$. We evaluate several target states, including <EOS>, "I don't know.", and "Hello.".

*Table 20.* Results on the MCF dataset under different $M_n$

| $M_n$ | Eff.↓ | Gen.↓ | Spe.↑ | PPL↓ |
|---|---|---|---|---|
| "<EOS>" | $0.40_{\pm 0.80}$ | $4.60_{\pm 2.24}$ | $14.90_{\pm 2.93}$ | $13.06_{\pm 0.18}$ |
| "I don't know." | $1.00_{\pm 1.34}$ | $5.70_{\pm 1.67}$ | $16.38_{\pm 3.50}$ | $13.15_{\pm 0.96}$ |
| "Hello." | $0.20_{\pm 0.60}$ | $4.80_{\pm 2.36}$ | $14.90_{\pm 2.70}$ | $13.21_{\pm 0.12}$ |

*Table 21.* Results on the ZsRE dataset under different $M_n$

| $M_n$ | Eff.↓ | Gen.↓ | Spe.↑ | PPL↓ |
|---|---|---|---|---|
| "<EOS>" | $27.85_{\pm 3.87}$ | $27.52_{\pm 3.87}$ | $27.73_{\pm 2.70}$ | $13.08_{\pm 0.06}$ |
| "I don't know." | $27.01_{\pm 2.91}$ | $26.81_{\pm 3.44}$ | $27.24_{\pm 2.70}$ | $13.06_{\pm 0.12}$ |
| "Hello." | $27.14_{\pm 2.75}$ | $27.50_{\pm 3.12}$ | $26.99_{\pm 2.34}$ | $13.09_{\pm 0.15}$ |

*Table 22.* Results on the MQUAKE dataset under different $M_n$

| $M_n$ | Eff.↓ | PPL↓ |
|---|---|---|
| "<EOS>" | $23.60_{\pm 3.07}$ | $13.11_{\pm 0.17}$ |
| "I don't know." | $22.66_{\pm 3.04}$ | $13.14_{\pm 0.10}$ |
| "Hello." | $24.51_{\pm 3.37}$ | $13.15_{\pm 0.16}$ |

