# OpenReview forum: "ZeroUnlearn: Few-Shot Knowledge Unlearning in Large Language Models"
_ICML.cc/2026/Conference — ICML 2026 regular_

### Official Review · Reviewer_771a · 2026-03-02

**Soundness:** 3
**Presentation:** 3
**Significance:** 3
**Originality:** 3
**Overall Recommendation:** 4
**Confidence:** 2

**Summary:**

This paper reformulates LLM machine unlearning as a precision knowledge remapping task, modifying specific MLP layers with three core constraints: orthogonalization of forget-set outputs, neutral target mapping for forget-set inputs, and mapping invariance for the retain set. Two approaches are proposed: a closed-form solution for few-shot unlearning and a gradient-based ZeroUnlearn-GD for multi-sample scenarios. Evaluations on MCF, ZsRE and MQUAKE show SOTA unlearning performance, and extensive experiments confirm the method achieves targeted knowledge removal while preserving LLMs’ core linguistic capabilities.
Overall, the authors address a pressing challenge of selective sensitive knowledge erasure without sacrificing LLM utility, and provide a model-editing-based framework to avoid the defects of traditional retraining/finetuning methods. The authors claim to study a pressing issue of inefficient and destructive LLM unlearning, where conventional methods either incur huge computational costs or cause catastrophic forgetting of related knowledge.

**Compliance With Llm Reviewing Policy:**

Affirmed.

**Final Justification:**

This manuscript designs a flexible framework with two complementary solutions, adapting to both few-shot and multi-sample unlearning scenarios. As for rebuttal, my concerns have been adequately addressed. I decide to maintain the initial score(weak accept).

**Key Questions For Authors:**

Please supplement the specific MLP layer indices/ranges adjusted for each evaluated model (Llama-3.2-3B-Instruct, Llama-3.1-8B-Instruct, Qwen3-4B) in the experimental details.
Clarify the sample selection criteria for the utility set and forget set: whether samples are randomly selected, the exact size of the utility set, and if there are screening rules
Confirm if all baseline methods are evaluated under the same few-shot/multi-sample settings as the proposed method. Provide a comparison of computational efficiency (training time, GPU memory consumption, number of parameter updates) between the proposed method and baselines.
Elaborate on the intrinsic relationship between the orthogonalization constraint and neutral output constraint for the forget set, and explain whether their effects can be verified separately via visualization methods.

**Limitations:**

The proposed unlearning method is only applicable to settings with explicitly defined forget sets and utility sets, and relies on the extraction of key representations from the forget set and the construction of the utility set to constrain parameter updates. It cannot handle unlearning tasks in real-world scenarios where the forget set is implicit, incomplete, or the utility set is difficult to construct, which limits its practical application scope.

**Strengths And Weaknesses:**

S1.Empirically verifies that updating MLP layers outperforms hidden/attention layers for unlearning, providing clear layer selection guidance for LLM knowledge editing.
2.Uses PCA visualization to clearly show the separation of forgotten/retained knowledge representations, complementing quantitative results with intuitive evidence for the method’s effectiveness.
3.The method proposed achieves near-perfect erasure (0% Eff.) without the severe PPL explosion and specificity collapse seen in GA/FT.
4.Designs a flexible framework with two complementary solutions, adapting to both few-shot and multi-sample unlearning scenarios.

W1.undefined notation on page 4 line 191, unclear y-axis labels in Figure 2, and no pseudocode for ZeroUnlearn-GD, hindering reproducibility and understanding.
2.The paper only conducts an ablation study on the neutral target state Mn​ and verifies its impact on unlearning performance, but lacks a comprehensive ablation analysis of the orthogonalization constraint and the mapping invariance constraint.
3.Although the paper focuses on few-shot machine unlearning and extends to multi-sample scenarios, it does not conduct a systematic analysis on how the number of forget set samples.
4.The experimental tables list all performance results of different methods but do not highlight the best and second-best performance with standardized formatting (e.g., bold for the best, underlined for the second-best). This increases the difficulty for readers to quickly compare and identify the advantages of the proposed methods over baselines, reducing the readability of the experimental results.

---

> ### Author Rebuttal · Authors · 2026-03-31
>
> Dear Reviewer 771a, thank you very much for your thoughtful comments and for recognizing the contribution of our work. To address your concerns as comprehensively as possible, we provide an anonymous experiment link (https://anonymous.4open.science/api/repo/ZeroUnlearn-9B87/file/1.pdf?v=cdb34a85) that contains the complete supplementary experimental results. All supplementary experiments were conducted on Llama-3.2.
>
> ---
> ## W1: On notation and figure clarity
> At the location you pointed out, we mistakenly wrote $\mathbf{M}_f$ as $\mathbf{H}_f$. Thank you for catching this; we will correct it in the revision. We will also clarify the y-axis labels in Figure 2 and add pseudocode for ZeroUnlearn-GD. Its pipeline is almost identical to ZeroUnlearn; the main difference is that ZeroUnlearn uses the closed-form update, while ZeroUnlearn-GD optimizes Eq. (12) directly.
>
> ## W2: On additional ablations
> The two constraints you mentioned correspond in our formulation to the orthogonal projection and the utility term. We therefore added two ablations: **w/o OP** replaces the orthogonal projection matrix with the identity matrix, and **w/o UT** removes the utility term.
>
> | ZsRE        | Eff. ↓       | Gen. ↓       | Spe. ↑       | PPL ↓        |
> | ----------- | :----------- | :----------- | :----------- | :----------- |
> | ZeroUnlearn | 27.85 ± 3.87 | 27.52 ± 3.87 | 27.73 ± 2.70 | 13.08 ± 0.06 |
> | w/o OP      | 28.45 ± 4.26 | 29.08 ± 4.72 | 27.99 ± 2.69 | 12.98 ± 0.09 |
> | w/o UT      | 23.65 ± 3.14 | 24.89 ± 3.25 | 25.28 ± 2.33 | 14.98 ± 0.35 |
>
> Replacing the orthogonal projection with the identity matrix noticeably weakens unlearning, especially on MQUAKE. Removing the utility term increases PPL by about 2 points, indicating clear utility degradation. Results on the remaining dataset are in the experiment link (Tables 10-11).
>
> ## W3: On analysis with forget-set size
> We added a systematic analysis over different forget-set sizes. ZeroUnlearn is stable in the few-shot setting, but as the number of forget samples grows to 500 and 1000, PPL rises noticeably, which further motivates ZeroUnlearn-GD for larger-scale settings.
>
> | MCF  | Eff. ↓      | Gen. ↓      | Spe. ↑       | PPL ↓        |
> | ---- | :---------- | :---------- | :----------- | :----------- |
> | 10   | 1.00 ± 3.00 | 7.00 ± 5.09 | 20.60 ± 4.86 | 13.10 ± 0.11 |
> | 25   | 0.80 ± 1.60 | 5.00 ± 2.86 | 17.52 ± 3.71 | 13.06 ± 0.13 |
> | 50   | 0.40 ± 0.80 | 4.60 ± 2.24 | 14.90 ± 2.93 | 13.06 ± 0.18 |
> | 100  | 0.50 ± 0.81 | 5.15 ± 1.78 | 14.72 ± 2.31 | 13.05 ± 0.19 |
> | 200  | 0.90 ± 0.70 | 5.90 ± 0.77 | 13.29 ± 0.86 | 13.25 ± 0.15 |
> | 500  | 3.25 ± 0.43 | 6.22 ± 0.25 | 12.53 ± 0.78 | 13.77 ± 0.38 |
> | 1000 | 6.73 ± 0.42 | 8.45 ± 0.66 | 12.20 ± 0.49 | 14.25 ± 0.41 |
>
> The experimental results on ZsRE and MQUAKE are provided in the experiment link (Table 7~9).
>
> ## W4: On  readability
> In the revision, we will adopt more standard table formatting, e.g., bold for the best and underline for the second-best, to improve readability. The supplementary tables in the experiment link already follow a more standardized format.
>
> ## Q1: MLP Layer Selection
> As described in Appendix E, for Llama-3.2 we select the layers with the strongest causal effect on the last token, namely Layers 16, 17, and 18. The causal effect computation follows ROME [1].
>
> ## Q2: Utility and Forget Set Construction
>
> For the few-shot setting, we use random seeds **1–10** to construct 10 data splits; for the multi-sample setting, we use random seed **1** for a single split. Following prior work, we randomly sample $10^5$ instances from Wikidata as the utility set.
> ## Q3: Consistency of Evaluation Settings
> Yes. All baselines are evaluated under the same few-shot or multi-sample settings as our method. We will make this more explicit in the experimental setup.
>
> ## Q4: Computational Efficiency
> We measured runtime and memory under varying forget-set sizes. The SVD step is very efficient: as the forget-set size increases from **10 to 1000**, runtime stays below **0.3 s** on MCF/ZsRE and **0.6 s** on MQUAKE, while memory only increases from **13.8 GB** to **14.1 GB**. The end-to-end editing cost grows roughly linearly with sample count, from **0.04 h** at 10 samples to **3.35–3.82 h** at 1000 samples, with memory remaining around **14.9–17.4 GB**. Thus, the closed-form update is not the bottleneck; the main cost comes from key/value extraction and layer-wise editing. Detailed trends are shown in Figures 1–4 of the experiment link.
>
> ## Q5: Relationship Between Orthogonality and Neutral Constraints
> We will clarify this relationship in the revision. Intuitively, the orthogonality constraint removes directions aligned with the original knowledge, while the neutral target constraint redirects the edited representation to a safe state. They serve complementary roles: one promotes erasure, the other promotes replacement.
>
> [1] Locating and editing factual associations in gpt. NeurIPS 2022

---

> > ### Author Rebuttal · Reviewer_771a · 2026-04-04
> >
> > Thanks for the detailed rebuttal.My concerns have been adequately addressed. I maintain the initial score(weak accept).

---

> > > ### Author Response · Authors · 2026-04-04
> > >
> > > Thank you for your feedback and for recognizing that our rebuttal has adequately addressed your concerns. We sincerely appreciate your time and your continued support of our work.

---

### Official Review · Reviewer_9DBK · 2026-03-08

**Soundness:** 1
**Presentation:** 2
**Significance:** 2
**Originality:** 2
**Overall Recommendation:** 4
**Confidence:** 3

**Summary:**

This paper formulate machine unlearning as a precise knowledge re-mapping problem via model editing and propose a few-shot unlearning framework, which re-maps the sensitive inputs to a neutral target state and removes their original representations by ensuring the unlearning process orthogonalizes the edited representations with respect to their original sensitive embeddings.

**Compliance With Llm Reviewing Policy:**

Affirmed.

**Final Justification:**

All of my concerns have been addressed. Based on the rebuttal, I am happy to raise my score to 4.

**Key Questions For Authors:**

1. If the paper aims to propose a new unlearning benchmark, it should be described in detail. For example, how to get the original model, what is the gold standard, how do you measure the knowledge memorization of original model, what is the evaluation metric and why it.  Otherwise, the proposed unlearning method should be evaluated against existing benchmarks [1-3] and compared with strong LLM unlearning baselines, such as NPO[4].
2. How do the authors ensure that the identified representation actually corresponds to the target knowledge related to the input?
3. Did the authors consider evaluating the robustness of ZeroUnlearn by asking reasoning-related questions to see whether the re-mapping mechanism can be bypassed?
4. Directly producing a neutral token may not be desirable for unlearning, since a simple detect-then-replace strategy (e.g., replacing the original output with "I don't know") could achieve a similar effect with lower computational cost. Why would ZeroUnlearn be preferable compared to this naive approach?

**Limitations:**

No. The paper includes a brief impact statement but could further discuss the limitations of editing-based unlearning and possible misuse scenarios.

**Strengths And Weaknesses:**

## Strengths

- The proposed unlearning method focuses on a dual objective: redirecting sensitive inputs to target tokens while minimizing the representational similarity between the updated state and the original knowledge.
- The paper provides a closed-form solution for the optimal transformation matrix through theoretical derivation.
- The experimental design is relatively fair and evaluates the method on two different LLM families (Llama & Qwen) and different model scales (3B & 8B).

## Weaknesses
_Soundness:_

1. The major weakness concerns the evaluation benchmark. The paper claims to propose an unlearning framework, but it remains unclear why typical and widely used unlearning benchmarks, such as MUSE-Bench[1], WMDP[2], or RWKU[3], are not used for evaluation. This limits direct comparison with previous unlearning methods.
2. The baseline comparison appears incomplete. Several typical LLM unlearning baselines, such as NPO [4] and SimNPO [5], are missing, which also only utilize the forget set.  The experimental conclusions can not fully reflect the effectiveness of ZeroUnlearn in the LLM unlearning setting. In particular, Gradient Ascent is known to be unstable for unlearning, while a more stable baseline such as NPO should be included.
3. In the Abstract, the authors claim that ZeroUnlearn outperforms existing baselines, which may be somewhat overstated. The empirical evidence does not fully support this claim. For example, in Table 1, ZeroUnlearn is not consistently the best method in terms of Spe.
4. Another concern is the robustness against various out-of-distribution attacks, such as reasoning attacks or role play attacks. The effectiveness of re-mapping target knowledge to a safe state should ideally be evaluated with a broader set of queries rather than only those from the training distribution. For example, the method may edit the relation "Paris -> France", but what happens if we ask a reasoning-style question such as: "Paris is the capital of which European country?" Can the model still recover the original mapping?


_Presentation_

- In Line 310 and 311, the datasets "MCF", "ZsRE" and "MQUAKE" should be mentioned with proper references or footnote links.
- In line 313, when introducing the first metric Efficacy, it is not numbered consistently with the subsequent metrics.


_Significance_

1. How can the authors ensure that the identified representation actually corresponds to the target knowledge related to the input?

2. In some cases, directly producing a neutral token may not be preferable, since users may expect a consistent response. What is the difference between ZeroUnlearn and a simpler approach that first detects sensitive information and then outputs "I don't know"? The reviewer believes that this naive baseline should also be included for comparison and concerns about the unlearning output.


[1] Shi, Weijia, et al. "Muse: Machine unlearning six-way evaluation for language models." _arXiv preprint arXiv:2407.06460_ (2024).

[2] Li, Nathaniel, et al. "The wmdp benchmark: Measuring and reducing malicious use with unlearning." _arXiv preprint arXiv:2403.03218_ (2024).

[3] Jin, Zhuoran, et al. "Rwku: Benchmarking real-world knowledge unlearning for large language models." _Advances in Neural Information Processing Systems_ 37 (2024): 98213-98263.

[4] Zhang, Ruiqi, et al. "Negative preference optimization: From catastrophic collapse to effective unlearning." _arXiv preprint arXiv:2404.05868_ (2024).

[5] Fan, Chongyu, et al. "Simplicity prevails: Rethinking negative preference optimization for llm unlearning." _arXiv preprint arXiv:2410.07163_ (2024).

---

> ### Author Rebuttal · Authors · 2026-03-31
>
> Dear Reviewer 9DBK, thank you for your valuable feedback. To address your concerns, a full set of supplementary experiments is available at: https://anonymous.4open.science/api/repo/ZeroUnlearn-9B87/file/1.pdf?v=cdb34a85 (all on Llama-3.2).
>
> ## W1&Q1: Insufficiency of the benchmark
>
> Our initial motivation comes from knowledge editing: since editing methods update specific factual knowledge, we study whether they can be extended to unlearning. Therefore, the original submission used standard editing benchmarks, including MCF, ZsRE, and MQUAKE.
>
> We agree that standard unlearning benchmarks are important. Following your suggestion, we added evaluation on RWKU and included NPO as a strong baseline. Since our method is parameter-efficient and only updates the down-projection matrices of three FFN layers, we implement two NPO variants for fairness: (i) NPO, which updates the same three layers, and (ii) NPO-full, which updates all parameters. We have updated the anonymous code repository to include both NPO variants and other baselines. Full RWKU results for all three levels are in the experiment link (Tables 1–3).
>
> | RWKU (level 1)     | Eff. ↓           | PPL ↓         |
> | ------------------ | :--------------- | :------------ |
> | NPO                | 39.07 ± 0.59     | 12.93 ± 0.01  |
> | NPO-full           | 35.91 ± 1.48     | 13.22 ± 0.14  |
> | ZeroUnlearn        | **35.70 ± 3.79** | 13.18 ± 0.18  |
> | **RWKU (level 2)** | **Eff. ↓**       | **PPL** **↓** |
> | NPO                | 38.98 ± 0.27     | 12.89 ± 0.01  |
> | NPO-full           | 35.56 ± 2.51     | 13.00 ± 0.09  |
> | ZeroUnlearn        | **35.47 ± 2.40** | 13.10 ± 0.14  |
> | **RWKU (level 3)** | **Eff. ↓**       | **PPL** **↓** |
> | NPO                | 39.71 ± 0.43     | 12.91 ± 0.01  |
> | NPO-full           | 37.66 ± 2.36     | 13.05 ± 0.08  |
> | ZeroUnlearn        | **34.01 ± 4.75** | 13.10 ± 0.18  |
>
> Across all RWKU levels,  ZeroUnlearn consistently outperforms NPO, and is competitive or superior to NPO-full (especially at level 3).
>
> ## W2: Insufficiency of the baseline
>
> As mentioned in W1,  we implemented and open-sourced two NPO variants, and evaluated them on MCF, ZsRE, and MQUAKE.
>
> | MCF         | Eff. ↓          | Gen. ↓          | Spe. ↑           | PPL ↓        |
> | ----------- | :-------------- | :-------------- | :--------------- | :----------- |
> | NPO         | 17.00 ± 4.40    | 18.30 ± 4.73    | 17.90 ± 3.30     | 12.92 ± 0.12 |
> | NPO-full    | 16.44 ± 4.69    | 18.22 ± 5.88    | **18.40 ± 4.05** | 13.03 ± 0.11 |
> | ZeroUnlearn | **0.40 ± 0.80** | **4.60 ± 2.24** | 14.90 ± 2.93     | 13.06 ± 0.18 |
>
> We observe that NPO is much weaker on MCF, the (s,r,o) dataset, while ZeroUnlearn performs well on both structured factual data and QA datasets. Results on the remaining dataset are in the experiment link (Tables 4–6).
>
> ## W3: On the Interpretation of Results in Table 1
>
> We apologize for the wording in the Abstract. Our claim is not uniform superiority, but a better trade-off between forgetting and utility. As shown in Table 1 of main text, although ZeroUnlearn is lower on Spe., it achieves larger gains in Gen. We will revise the Abstract to reflect this more accurately (e.g., competitive overall performance).
>
> ## W4&Q3: On robustness of ZeroUnlearn
>
> Our evaluation includes robustness to paraphrased queries via the Gen. metric (e.g., ZsRE uses semantically equivalent rephrasings). We agree broader OOD settings (e.g., reasoning attacks) are important but not fully covered, and will clarify this limitation.
>
> ## W5: Presentation
>
> Thank you for pointing these out. We will fix them in the revision:
>
> - add explicit citations for MCF, ZsRE, MQUAKE in the main text.
>
> - fix metric numbering inconsistencies (Efficacy).
>
>
>
> ## W6&Q2: How we know the identified representation
>
> Following ROME [1], we locate target representations via causal tracing, using the last subject subtoken. Our AIE analysis (Appendix E) shows this token is the most causally relevant for factual retrieval, supporting our design.
> ## W7&Q4: Comparison to a simple method
>
> We consider two interpretations.
> (i) External detect-then-replace: this modifies the inference pipeline. Our goal is an end-to-end edited model that removes sensitive knowledge directly, without runtime intervention.
> (ii) Using “I don’t know.” as target:
>
> | MCF               | Eff. $\downarrow$ | Gen. $\downarrow$ | Spe. $\uparrow$ | PPL $\downarrow$ |
> | ----------------- | ----------------- | ----------------- | --------------- | ---------------- |
> | `"<EOS>"`         | 0.40 ± 0.80       | 4.60 ± 2.24       | 14.90 ± 2.93    | 13.06 ± 0.18     |
> | `"I don't know."` | 1.00 ± 1.34       | 5.70 ± 1.67       | 16.38 ± 3.50    | 13.15 ± 0.96     |
>
> It can be observed that setting the target to “I don’t know.” yields results comparable to those obtained with `<EOS>`. Results on the remaining dataset are in the experiment link (Tables 12-14).
>
> [1] Locating and editing factual associations in gpt. NeurIPS 2022

---

> > ### Author Rebuttal · Reviewer_9DBK · 2026-04-03
> >
> > Thanks for the detailed rebuttal. I appreciate the authors' responses. Most of my concerns have been addressed. The only remaining point is about reasoning-style attacks, which are not fully explored. I think this is acceptable given that the paper already includes basic attack settings and demonstrates preliminary robustness.
> >
> > Based on the rebuttal, I am happy to raise my score to 4.

---

> > > ### Author Response · Authors · 2026-04-03
> > >
> > > Thank you again for your thoughtful follow-up and for indicating that you are happy to raise your score. We sincerely appreciate your positive assessment and your recognition that most of our concerns have been addressed. We just want to kindly note that the score currently visible in the system still appears unchanged, so we are not sure whether the update has already gone through. It may simply be a system synchronization issue, but we think it might be helpful to mention it.
> > >
> > > Regarding the remaining point on reasoning-style attacks, we fully agree that this is an important direction. In preparing the rebuttal, we looked for a standardized benchmark that specifically targets reasoning-style attack prompts in the factual editing setting, but were not able to identify one that is commonly adopted for this purpose. As a partial proxy, we additionally examine the **reference score (RS)** on the MCF dataset. This metric is introduced in ROME [1] to measure whether the model’s generations remain semantically consistent with texts associated with the target. For example, consider a forget sample in the ($s$,$r$,$o$) form: *“What does Dianne Reeves play? They play jazz.”*, where *"jazz"* is the target object ($o$) to be unlearned. The MCF dataset further provides a set of subject-related prefixes for *"Dianne Reeves"*, such as *“Dianne Reeves’s greatest artistic work is”*. These prefixes introduce additional contextual cues related to the original ($s$,$r$) pair. For instance, *"artistic work"* may make it easier for the model to infer and generate content associated with the target object $o$ during open-ended generation.  Following ROME, for each subject-related prefix we let the model generate a continuation of **100 tokens**, and then measure the semantic consistency between the generated text and reference texts associated with the target property $o$, using cosine similarity over unigram TF-IDF vectors. The reference texts are taken from the `attribute_snippets.json` file in https://rome.baulab.info/data/dsets.
> > >
> > > Intuitively, this metric helps assess whether the model can still recover target-related semantic content under freer and more indirect generation settings, rather than only through explicit prompts that directly query the forgotten fact. We report the results of AlphaEdit, the strongest knowledge editing baseline in our experiments, together with NPO and NPO-full:
> > >
> > > | Method      | Reference Score  |
> > > | ----------- | :--------------- |
> > > | AlphaEdit   | 30.57 ± 1.55     |
> > > | NPO         | 32.50 ± 1.59     |
> > > | NPO-full    | 33.11 ± 1.67     |
> > > | ZeroUnlearn | **28.90** ± 2.06 |
> > >
> > > As shown above, ZeroUnlearn achieves the lowest reference score among all compared methods. Since a lower RS indicates weaker semantic consistency with texts expressing the target property, this result suggests that ZeroUnlearn is more effective at suppressing target-related semantic content even under open-ended generation with indirect subject-related prefixes. At the same time, we agree that this is only partial evidence and **not** a substitute for a dedicated benchmark specifically designed for reasoning-style attacks.
> > >
> > > [1] Locating and editing factual associations in gpt. NeurIPS 2022

---

### Official Review · Reviewer_bajm · 2026-03-09

**Soundness:** 4
**Presentation:** 3
**Significance:** 3
**Originality:** 4
**Overall Recommendation:** 4
**Confidence:** 4

**Summary:**

This paper reframes LLM machine unlearning as a knowledge re-mapping problem via model editing. It proposes to overwrite sensitive knowledge by mapping forget-set inputs to a neutral EOS token representation while enforcing orthogonality between the updated and original representations through a multiplicative weight update with a closed-form solution. For larger batch unlearning, a gradient-based variant (ZeroUnlearn-GD) is introduced.

**Compliance With Llm Reviewing Policy:**

Affirmed.

**Final Justification:**

The authors have generally addressed my comments, and with their revisions, the paper can be a positive paper.

**Key Questions For Authors:**

Weaknesses

**Limitations:**

yes

**Strengths And Weaknesses:**

## Strengths

- **Clean math.** The three-term objective is well-motivated, the null-space projection has nice geometric intuition, and the closed-form solution for the few-shot case is elegant.
- **Decent experimental coverage.** Three datasets, three models, both few-shot and batch settings, downstream tasks, PCA visualizations, ablations.
- **Better than most baselines on efficacy.** ZeroUnlearn does improve over ROME and MEMIT, though the margins over AlphaEdit are more modest.

## Weaknesses

- **Multiplicative → additive switch is never justified.** The whole pitch is that multiplicative updates are better than additive ones. Then Section 5 quietly switches to additive for multi-sample. If multiplicative is so great, why abandon it? This contradicts the paper's own argument.

- **SVD cost is ignored.** They analyze the O(d⁶) bottleneck of the Kronecker inversion but never discuss the cost of the SVDs they run per layer. No wall-clock comparisons anywhere.

- **They do use a retain set.** The paper claims only the forget set is needed, but they sample 10⁵ Wikidata entries as a "utility set" — that's a retain set by another name. This makes baseline comparisons unfair since GA gets no retain data. Either give all methods the same utility set or stop claiming you don't use one.

- **Layer selection is ad hoc.** Causal tracing points to early layers, but editing those breaks the model, so they pick middle layers instead. This is a post-hoc fix that requires manual tuning per architecture and undermines the "surgical" claims.

- **"Few-shot" is a stretch.** 50 forget samples with d > 3000 isn't really few-shot in any meaningful sense. The null space being high-dimensional when n ≪ d is expected, not impressive. And the 10⁵ utility samples further undercut this framing.

- **Specificity drops are worse than presented.** ~40% relative drop in the multi-sample setting on Llama-3.2. Saying "it's better than GA" is a low bar.

- **Narrow evaluation scope.** Only clean (subject, relation, object) triples. Real unlearning needs (concepts, behaviors, GDPR compliance) are much messier.

- **Missing baselines.** Since the method effectively uses a retain set, comparison should include retain-set-aware unlearning methods. Notable omissions include NPO [1], GradDiff [2], and PDU [3]. Ideally the authors would provide their Wikidata utility set to all baselines for a fair comparison, and also evaluate on the TOFU benchmark [2] that the community has converged on.

## References

[1] Zhang et al., "Negative Preference Optimization: From Catastrophic Collapse to Effective Unlearning"

[2] Maini et al., "TOFU: A Task of Fictitious Unlearning for LLMs"

[3] Entesari et al., "Constrained Entropic Unlearning: A Primal-Dual Framework for Large Language Models"

---

> ### Author Rebuttal · Authors · 2026-03-31
>
> Dear Reviewer bajm, thank you for your thoughtful and constructive feedback. We respond to each concern below. A full set of supplementary experiments is available at: https://anonymous.4open.science/api/repo/ZeroUnlearn-9B87/file/1.pdf?v=cdb34a85 (all on Llama-3.2).
>
> ------
>
> ## W1: On the additive switch
>
> We do not claim multiplicative updates are universally better, but that they are particularly suitable in the few-shot regime, where $\textbf{M}_f$ is low-rank and the null space is large. In multi-sample settings, multiplicative updates can cause rank collapse and hurt generalization. We therefore added the following comparison at 1000 edits:
>
> | MCF                   | Eff. ↓      | Gen. ↓      | Spe. ↑       | PPL ↓        |
> | --------------------- | :---------- | :---------- | :----------- | :----------- |
> | ZeroUnlearn (50)      | 0.40 ± 0.80 | 4.60 ± 2.24 | 14.90 ± 2.93 | 13.06 ± 0.18 |
> | ZeroUnlearn (1000)    | 6.73 ± 0.42 | 8.45 ± 0.66 | 12.20 ± 0.49 | 14.25 ± 0.41 |
> | ZeroUnlearn-GD (1000) | 0.00        | 5.10        | 12.41        | 13.05        |
>
> The key difference is PPL: scaling multiplicative updates raises PPL (13.06 to 14.25), while the additive GD variant remains stable (13.05). This supports our design: multiplicative for few-shot, additive for large-scale stability.
>
> ## W2: On the cost of SVD and practical runtime
>
> We measured runtime and memory. SVD is lightweight: <0.3s (MCF/ZsRE) and <0.6s (MQUAKE) for 10–1000 samples, with minimal memory increase (13.8 to 14.1GB). In contrast, total runtime grows from 0.04 to 3.5h, dominated by key/value extraction and layer-wise updates. Thus, SVD is not the bottleneck. The corresponding line plots are in Figures 1–4 of the experiment link.
>
> ## W3: On the utility set
>
> By “only the forget set,” we mean no task-specific retain dataset with labels is required. The 10⁵ Wikidata samples serve as a generic preservation resource, consistent with prior editing methods (e.g., MEMIT[1], AlphaEdit[2]). We agree this should be clarified and will revise accordingly.
>
> ## W4: On layer selection
>
> Layer selection is guided by causal tracing (following ROME[3]), which identifies a contiguous region responsible for factual recall. While early-layer editing may harm utility, we select layers balancing causal relevance and preservation. We will clarify this and soften the “surgical” claim.
>
> ## W5: On the few-shot characterization
>
> In LLM editing and targeted factual unlearning, editing on 50 forget samples is still highly data-efficient compared to methods requiring broader re-optimization or large retain data. Our point is not that $n\ll d$ is surprising, but that this is exactly the regime where the multiplicative null-space formulation is most suitable. We agree the presentation should be more precise. In the revision, we will clarify that “few-shot” refers to the number of forget samples, while the utility set is a generic preservation resource from prior work.
>
> ## W6: Specificity drops in the multi-sample setting
>
> We agree the current wording may understate this trade-off. In the multi-sample setting on Llama-3.2, specificity drops noticeably relative to the base model, and this should be stated more clearly. Our claim is not that locality is fully preserved, but that our method outperforms strong baselines (e.g., GA, FT), which cause more severe perplexity collapse. We will revise accordingly.
>
> ## W7: Narrow evaluation scope
>
> Among our datasets, only MCF is strictly triple-structured. ZsRE, MQUAKE, and the newly added RWKU [4] are all QA-style datasets, and ZeroUnlearn still shows strong unlearning performance on them.
>
> ## W8: Insufficiency of the baseline
>
> In the revision, we extend the comparison to include NPO [4], a strong unlearning baseline, and evaluate on RWKU. To ensure fairness, we implement both a parameter-efficient NPO variant that updates the same three FFN layers as our method and a full fine-tuning version (NPO-full).
>
> | **RWKU (level 3)** | **Eff. ↓**       | **PPL** **↓** |
> | ------------------ | :--------------- | :------------ |
> | NPO                | 39.71 ± 0.43     | 12.91 ± 0.01  |
> | NPO-full           | 37.66 ± 2.36     | 13.05 ± 0.08  |
> | ZeroUnlearn        | **34.01 ± 4.75** | 13.10 ± 0.18  |
>
> The complete results on all three RWKU levels are provided in the experiment link (Tables 1-3). Across all three RWKU levels, ZeroUnlearn consistently outperforms NPO, and also slightly or clearly outperforms NPO-full, especially at level 3. Results of NPO on the other three datasets are provided in the experiment link (Tables 4–6).
>
> [1] Mass-editing memory in a transformer. ICLR 2023
>
> [2] Alphaedit: Null-space constrained knowledge editing for language models. ICLR 2025
>
> [3] Locating and editing factual associations in gpt. NeurIPS 2022
>
> [4] Rwku: Benchmarking real-world knowledge unlearning for large language models. NeurIPS 2024
>
> [5] Negative preference optimization: From catastrophic collapse to effective unlearning. COLM 2024

---

> > ### Author Rebuttal · Reviewer_bajm · 2026-04-03
> >
> > The authors have generally addressed my comments and with their revisions, the paper can be a positive paper.

---

> > > ### Author Response · Authors · 2026-04-04
> > >
> > > Thank you for your encouraging comments and for acknowledging that our revisions have addressed your concerns. We sincerely appreciate your thoughtful feedback and support, which have helped strengthen the paper.

---

### Official Review · Reviewer_ERDD · 2026-03-12

**Soundness:** 3
**Presentation:** 3
**Significance:** 3
**Originality:** 3
**Overall Recommendation:** 4
**Confidence:** 4

**Summary:**

This paper studies the problem of machine unlearning in LLMs, aiming to remove specific factual knowledge while preserving the model’s overall capabilities. The authors reformulate unlearning as a knowledge remapping problem and propose ZeroUnlearn, a framework that edits model parameters to redirect sensitive inputs to a neutral target state. The method is built upon a knowledge editing formulation where factual associations are modeled as mappings between internal representations in an MLP layer, and the update is constrained so that the modified representations become orthogonal to the original sensitive representations. The paper derives a closed-form multiplicative parameter update for few-shot unlearning and further introduces a gradient-based variant for multi-sample scenarios. Experiments on several LLMs and benchmark datasets evaluate the approach using multiple metrics related to unlearning effectiveness, generalization, specificity, and language modeling performance.

**Compliance With Llm Reviewing Policy:**

Affirmed.

**Final Justification:**

I checked the reply. My questions are answered properly. I do not have further questions.

**Key Questions For Authors:**

1. The method introduces a large utility set constructed from approximately $10^5$ Wikidata samples to enforce preservation constraints. Could the authors clarify whether the baselines were provided with comparable information when performing editing or unlearning? Additionally, it would be helpful to include an ablation showing the performance of ZeroUnlearn without this utility set, to better understand how much it contributes to the reported results. Such clarification would help assess the fairness of the comparisons.
2. The proposed method redirects sensitive inputs to a neutral target corresponding to the `<EOS>` token. Could the authors discuss whether the observed unlearning behavior depends on this specific choice? For example, would the results remain similar if other neutral targets were used, or if the model were encouraged to produce a generic refusal response instead? Additional analysis would help clarify whether the method performs genuine knowledge removal rather than output suppression.
3. The paper motivates the work partly from privacy and safety considerations. However, the evaluation mainly focuses on QA-style metrics. Have the authors considered evaluating the method using more direct privacy-oriented metrics, such as membership inference or data extraction attacks? Including such evaluations could strengthen the claims regarding privacy and sensitive information removal.
4. The paper provides a complexity discussion for the multi-sample closed-form and motivates the GD approximation. However, it would still be helpful to include: (i) The practical runtime/memory breakdown for the few-shot closed-form. (ii) How the total cost scales with the number of edited layers and forget-set size in practice.

**Limitations:**

Yes

**Strengths And Weaknesses:**

Pros:

 - The paper addresses the practically important problem of machine unlearning in LLMs, which is relevant for privacy, safety, and model maintenance. The authors provide a clear motivation by highlighting the limitations of existing retraining and fine-tuning approaches, and reformulate unlearning as a knowledge remapping problem through model editing. This framing is intuitive and provides a conceptual basis for the proposed method.

 - The paper presents a well-structured derivation of the proposed method, formulating the objective with terms that enforce forgetting, preservation of unrelated knowledge, and representational orthogonality. Under a multiplicative update parameterization, the authors derive a closed-form solution that enables efficient one-step updates in few-shot settings. The overall methodological pipeline is clearly explained from objective formulation to the resulting update rule.

 - The experimental section evaluates the method on several LLMs and benchmarks using multiple metrics that capture different aspects of unlearning performance, including effectiveness, generalization, specificity, and language modeling quality. In addition to quantitative metrics, the paper also includes representation-level analysis to study how edited representations change after unlearning, providing additional insight into the behavior of the proposed method.

Cons:

 - The method uses a utility set of $10^5$ samples (described as coming from Wikidata / Wikipedia) to enforce preservation constraints. This effectively introduces additional auxiliary data beyond the forget set during editing. It would be helpful to clarify whether the baselines are provided with comparable information, or to include ablations without this utility set to ensure a fair comparison.

 - The proposed method redirects sensitive inputs to a neutral target representation corresponding to the `<EOS>` token. Although the paper argues that the method “does not merely suppress the original output” but learns a remapping to a predefined null state, in practice mapping to `<EOS>` may still resemble a refusal/suppression behavior at generation time, rather than removing the underlying knowledge. Additional analysis with alternative neutral targets or generation behavior would help clarify this point.

 - The paper motivates the work partly from privacy and safety considerations, but the evaluation mainly focuses on QA-style metrics. More direct privacy-oriented evaluations, such as membership inference or data extraction attacks, would better support the claims regarding privacy and sensitive information removal. Including such analyses could strengthen the empirical validation of the method.

---

> ### Author Rebuttal · Authors · 2026-03-31
>
> Dear Reviewer ERDD, thank you very much for your thoughtful comments and for recognizing the contribution of our work. Here are our responses to your comments. To address your concerns as comprehensively as possible, we provide an anonymous experiment link (https://anonymous.4open.science/api/repo/ZeroUnlearn-9B87/file/1.pdf?v=cdb34a85) that contains the complete supplementary experimental results. All supplementary experiments were conducted on Llama-3.2.
>
> ---
> ## W1&Q1: Fairness of comparison
>
> The use of $10^5$ Wikidata samples to preserve general capability was introduced by MEMIT [1] and has been widely adopted in the editing literature. We follow this standard practice to construct the utility term. We also add an ablation of the utility term below:
>
> | ZsRE        | Eff. $\downarrow$ | Gen. $\downarrow$ | Spe. $\uparrow$ | PPL $\downarrow$ |
> | ----------- | :---------------- | :---------------- | :-------------- | :--------------- |
> | ZeroUnlearn | 27.85 ± 3.87      | 27.52 ± 3.87      | 27.73 ± 2.70    | 13.08 ± 0.06     |
> | w/o UT      | 23.65 ± 3.14      | 24.89 ± 3.25      | 25.28 ± 2.33    | 14.98 ± 0.35     |
>
> Results on other datasets are provided in the experiment link (Tables 10–11). Removing the utility term increases PPL by about **2 points**, indicating a clear drop in general capability.
> ## W2&Q2: On other neutral targets
> To test whether the behavior depends on the specific choice of `<EOS>`, we additionally evaluate two alternative neutral targets, _“I don’t know”_ and _“Hello”_.
>
> | MCF               | Eff. $\downarrow$     | Gen. $\downarrow$     | Spe. $\uparrow$     | PPL $\downarrow$     |
> | ----------------- | :-------------------- | :-------------------- | :------------------ | :------------------- |
> | `"<EOS>"`         | 0.40 ± 0.80           | 4.60 ± 2.24           | 14.90 ± 2.93        | 13.06 ± 0.18         |
> | `"I don't know."` | 1.00 ± 1.34           | 5.70 ± 1.67           | 16.38 ± 3.50        | 13.15 ± 0.96         |
> | `"Hello."`        | 0.20 ± 0.60           | 4.80 ± 2.36           | 14.90 ± 2.70        | 13.21 ± 0.12         |
> | **ZsRE**          | **Eff. $\downarrow$** | **Gen. $\downarrow$** | **Spe. $\uparrow$** | **PPL $\downarrow$** |
> | `"<EOS>"`         | 27.85 ± 3.87          | 27.52 ± 3.87          | 27.73 ± 2.70        | 13.08 ± 0.06         |
> | `"I don't know."` | 27.01 ± 2.91          | 26.81 ± 3.44          | 27.24 ± 2.70        | 13.06 ± 0.12         |
> | `"Hello."`        | 27.14 ± 2.75          | 27.50 ± 3.12          | 26.99 ± 2.34        | 13.09 ± 0.15         |
>
> Results on the remaining dataset are in the experiment link (Tables 12–14). Across datasets, performance is broadly comparable to `<EOS>`, suggesting that ZeroUnlearn does not rely on a particular surface-form target, but instead remaps the targeted knowledge to a neutral state more generally.
> ## W3&Q3: Privacy and Safety Claims
>
> Thank you for this suggestion. We agree that privacy-oriented evaluation would strengthen claims about sensitive information removal. Beyond QA-style evaluation, however, we also include MCF, which measures factual associations at the structured (s,r,o) level rather than only through QA prompts. This provides a complementary view of whether the target fact itself has been removed.
>
> More broadly, current LLM unlearning evaluations mainly focus on QA-style and knowledge-removal benchmarks [2,3]. To the best of our knowledge, membership inference and data extraction attacks are not yet standard protocols in this line of work. We agree these are valuable directions and will clarify this limitation in the revision.
>
> ## Q4: On the cost of ZeroUnlearn
> To complement the theoretical complexity discussion, we additionally measured the practical runtime and memory usage of the few-shot closed-form update across different forget-set sizes. The results show that the SVD step itself is very lightweight: even when the forget-set size increases from 10 to 1000, the average SVD time remains below **0.3 seconds** on MCF/ZsRE and below **0.6 seconds** on MQUAKE, while the corresponding memory usage only increases modestly from about **13.8 GB** to **14.1 GB**.
>
> We also report the end-to-end cost of the full editing procedure. As expected, runtime grows approximately linearly with forget-set size, from about **0.04 h** at 10 samples to **3.35–3.82 h** at 1000 samples across datasets. Total memory remains stable at roughly **14.9–17.4 GB**. These results suggest that the closed-form update is not the practical bottleneck; the main cost comes from key/value extraction and layer-wise editing. Corresponding line plots are provided in **Figures 1–4** of the experiment link.
>
> [1] Mass-editing memory in a transformer. ICLR 2023
>
> [2] Large language model unlearning. NeurIPS 2024
>
> [3] Machine unlearning of pre-trained large language models. ACL 2024

---

> > ### Author Rebuttal · Reviewer_ERDD · 2026-04-03
> >
> > Thanks for the reply.

---

> > > ### Author Response · Authors · 2026-04-04
> > >
> > > Thank you for your positive assessment and for confirming that your concerns have been fully addressed. We sincerely appreciate your time, careful review, and constructive feedback, which helped improve the quality of our manuscript.

---

### Decision · Program_Chairs · 2026-04-30

**Decision:**

Accept (regular)

**Comment:**

This paper introduces ZeroUnlearn, a few-shot unlearning framework that treats machine unlearning as a knowledge re-mapping problem using model editing. The main idea centers on a three-part objective: it forces the model to forget by making representations orthogonal, maps sensitive inputs to neutral targets, and keeps unrelated knowledge intact with a utility term. In the few-shot case, it updates parameters in closed form. For scenarios with more samples, it switches to a gradient-based update. This approach tackles real problems with current optimization-based unlearning methods, especially their sensitivity to hyperparameters and the risk of destroying unrelated model knowledge.

All four reviewers landed on weak accept. The authors addressed every major concern in their response, backing it up with new experiments. Reviewer bajm had questions about why the update goes from multiplicative to additive with more samples, the not-yet-disclosed use of a utility set, missing baselines, and choices about which layer to edit. The authors explained each point. They justified the design change, explained that using a utility set follows prior work like MEMIT and AlphaEdit, and added strong baselines (NPO and NPO-full) in all experiments. Reviewer 9DBK started out skeptical, giving the paper a low score, but after the rebuttal (which added RWKU evaluation, compared with NPO, included a sensitivity analysis of neutral targets, and added a reference score to check reasoning robustness), bumped the soundness score to 4. Reviewers ERDD and 771a also said their concerns had been addressed.

Some open questions remain, but they’re fair limitations, not core problems: the method addresses robustness to reasoning-based attacks (beyond paraphrasing), and it depends on having clear forget and utility sets, which won't always be at hand. Privacy attacks like membership inference aren’t covered yet. Still, these limits just mark out the scope—they don’t undermine the core advance. ZeroUnlearn gives a technically sound and well-tested approach to targeted factual unlearning that performs well against strong baselines. So we recommend acceptance.